# Geometry of Reason: Spectral Signatures of Valid Mathematical Reasoning

**Valentin Noël** [1]

## Abstract

Verifying whether a language model is genuinely reasoning or pattern-matching remains an open problem: learned verifiers are expensive, and output-based heuristics are brittle. We show that valid mathematical reasoning induces a measurable, training-free spectral signature in transformer attention. By treating each attention matrix as a weighted token graph, we extract four diagnostics: Fiedler value, High-Frequency Energy Ratio (HFER), spectral entropy, and smoothness, that require no learned parameters. Experiments across seven models from four architectural families yield effect sizes up to Cohen's $d = 3.30$ ($p < 10^{-116}$), enabling 85–96% single-threshold classification accuracy. Two findings sharpen the interpretation. First, *Platonic validity*: the spectral signal tracks logical coherence rather than compiler acceptance, proofs rejected for timeouts or missing imports are correctly classified as valid, a distinction confirmed by a manual audit ($\kappa = 0.82$, $n = 51$). Second, *architectural determinism*: Sliding Window Attention shifts the discriminative feature from HFER to smoothness ($d = 2.09$, $p < 10^{-48}$), showing that attention design governs which spectral channel encodes reasoning quality. Causal ablation confirms the signature traces induction-head circuits. The method generalises to informal chain-of-thought ($d = 0.78$, $p < 10^{-3}$), and in proof search, HFER reranking improves Best-of-16 Pass@1 by +4.4–6.6%, matching 98% of the AUC of fully supervised probes with zero labels. Spectral graph analysis is a principled, architecture-aware primitive for reasoning verification. [*]

[1]Devoteam, Levallois-Perret, France. Correspondence to: Valentin Noël <valentin.noel@devoteam.com>.

*Proceedings of the $43^{rd}$ International Conference on Machine Learning*, Seoul, South Korea. PMLR 306, 2026. Copyright 2026 by the author(s).

[*] 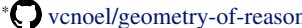 vcnoel/geometry-of-reason

## 1. Introduction

The remarkable performance of large language models (LLMs) on mathematical reasoning tasks (Lewkowycz et al., 2022; Trinh et al., 2024; Chervonyi et al., 2025; Azerbayev et al., 2023) has intensified interest in understanding and verifying the computational mechanisms underlying their outputs. When a model generates a mathematical proof, practitioners face a fundamental, epistemological, challenge: determining whether the output reflects genuine logical reasoning or sophisticated pattern matching that produces plausible-looking but potentially flawed arguments. Recent evaluations reinforce this concern: even frontier models achieve below 25% on olympiad-level proofs (Petrov et al., 2025), suggesting a "reasoning illusion" where success may stem from pattern matching rather than genuine insight (Kuang et al., 2025). This challenge is particularly acute in high-stakes applications such as automated theorem proving (Polu & Sutskever, 2020; Yang et al., 2024; Ospanov et al., 2025), mathematical education (Welleck et al., 2022), and scientific discovery (Romera-Paredes et al., 2024), where undetected reasoning errors can propagate with significant consequences.

Current approaches to reasoning verification fall into two broad categories, each with substantial limitations. Output-based verification relies on formal proof assistants such as Lean (de Moura & Ullrich, 2021), Coq (Bertot & Castéran, 2013), or Isabelle (Paulson, 1994) to check whether generated proofs compile successfully. While sound, this approach conflates logical validity with syntactic acceptability: proofs may be rejected due to timeout constraints, missing library imports, version incompatibilities, or formatting issues rather than genuine logical errors. Conversely, proofs with subtle semantic errors may pass compilation if they exploit gaps in type checking or axiom systems. Learned verification trains classifiers on model internals (Azaria & Mitchell, 2023; Burns et al., 2023; Marks & Tegmark, 2024) or output features (Lightman et al., 2023; Cobbe et al., 2021) to predict correctness. Recent work demonstrates that internal representations contain rich signals for hallucination detection (Chen et al., 2024; Zhang et al., 2025a; Healy et al., 2026), yet these methods require substantial labeled data, may not generalize across model architectures, and risk learning spurious correlations rather than fundamental properties of valid reasoning.

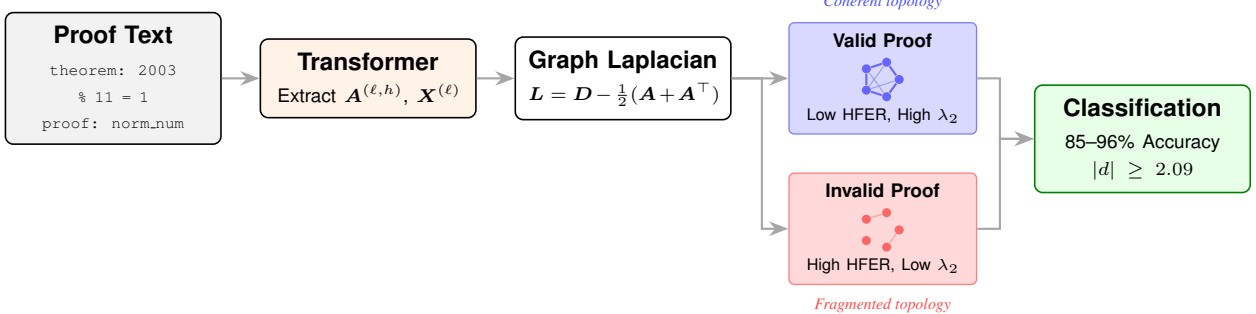

*Figure 1.* **Method Overview.** Spectral analysis of attention graphs enables training-free validity classification ($d$ up to 3.30).

We propose an alternative paradigm grounded in spectral graph theory. Our central insight is that self-attention induces a dynamic weighted graph over tokens, where edge weights correspond to attention scores. This perspective aligns with recent work on attention graphs for mechanistic interpretability (El et al., 2025; Rai et al., 2024). The spectral properties of this graph, eigenvalues and eigenvectors of its Laplacian, encode global structural information about information routing.

This hypothesis draws motivation from spectral graph theory (Chung, 1997) and neural manifold geometry (Chung et al., 2018; Cohen et al., 2020). Recent work establishes spectral theories connecting representation geometry to neural prediction (Canatar et al., 2023), showing that manifold capacity, how efficiently neural populations encode task-relevant structure, is captured by geometric and spectral properties (Chou et al., 2024; 2025). Findings that attention can be made vastly sparser while preserving capability (Draye et al., 2025) suggest essential structure may reside in low-dimensional spectral features. We thus expect valid proofs to induce smoother, well-connected attention graphs, and invalid reasoning to exhibit spectral irregularity.

**Contributions.** We make the following contributions:

1. We introduce a **training-free framework** for reasoning validity detection based on spectral analysis of attention graphs, achieving 82.8–85.9% accuracy under nested cross-validation and up to 95.6% with calibrated thresholds (Sections 3 and 4).

2. We demonstrate **cross-architecture universality** across seven models from four families with $|d| \geq 2.09$ and $p_{MW} < 10^{-47}$ in all cases, and show robustness across all difficulty strata ($d \geq 1.31$ for complex proofs, Section 4).

3. We discover **Platonic validity**: the spectral method detects logical coherence rather than compiler acceptance, identifying mathematically sound proofs that formal systems reject on technical grounds (Section 5.2).

## 2. Related Work

**Mechanistic Interpretability.** Understanding the internal computations of transformers has been a central goal of interpretability research. Foundational work by Elhage et al. (2021) introduced mathematical frameworks for analyzing transformer circuits, while Olsson et al. (2022) identified "induction heads" as key mechanisms for in-context learning. Subsequent work has analyzed attention patterns in arithmetic (Nanda et al., 2023; Hanna et al., 2023), factual recall (Meng et al., 2022; Geva et al., 2023), and reasoning tasks (Stolfo et al., 2023; Liu et al., 2022). Recent surveys (Rai et al., 2024) and tutorials systematize these techniques, while new methods enable attention head intervention for causal analysis (Kadem & Zheng, 2026) and sparse autoencoder-based feature extraction. Our work complements these approaches by analyzing the global geometric structure of attention rather than identifying specific circuits.

**Probing and Representation Analysis.** Linear probes have been extensively used to extract linguistic (Hewitt & Manning, 2019; Tenney et al., 2019), semantic (Ettinger, 2020; Li et al., 2021), and factual (Petroni et al., 2019) information from transformer representations. Recent work has probed for truthfulness (Azaria & Mitchell, 2023; Marks & Tegmark, 2024), uncertainty (Kadavath et al., 2022; Kuhn et al., 2023), and reasoning capabilities (Saparov et al., 2023; Dziri et al., 2024). Notably, 2025 work demonstrates that internal representations contain discriminative signals for hallucination detection (Chen et al., 2024; Zhang et al., 2025a;c), with attention outputs often providing stronger signals than other internal representations. However, concerns about probe reliability (Belinkov, 2022; Hewitt & Liang, 2019) motivate training-free alternatives. Our spectral approach requires no learned parameters and operates on attention structure rather than hidden states.

**Graph Signal Processing on Neural Networks.** The application of spectral graph theory to neural networks has a rich history, beginning with spectral graph convolutions (Bruna et al., 2013; Defferrard et al., 2016; Kipf &

Welling, 2017). Recent work analyzes transformers through graph-theoretic lenses, studying over-smoothing (Shi et al., 2022; Rusch et al., 2023) and designing spectral attention mechanisms (Kreuzer et al., 2021; Bo et al., 2023). El et al. (2025) introduce Attention Graphs for mechanistic interpretability, establishing mathematical equivalences between message passing and self-attention. Draye et al. (2025) show that attention can be made orders of magnitude sparser while preserving capability, suggesting redundancy exploitable by spectral methods. Our work extends this line by extracting multiple spectral diagnostics correlated with reasoning validity.

**LLM Verification and Hallucination Detection.** Verifying LLM outputs has received substantial attention. Process-based reward models (Lightman et al., 2023; Uesato et al., 2022) train verifiers on step-level annotations, while self-consistency methods (Wang et al., 2023) leverage sampling diversity. For hallucination detection, recent approaches leverage internal representations (Healy et al., 2026), eigenvalue analysis of hidden states and latent probing (Bhatnagar et al., 2026). Our method differs fundamentally: we require no training, no sampling, and operate on attention geometry rather than output content or hidden state magnitudes.

**Neural Theorem Proving.** LLMs have achieved impressive results in formal mathematics. Recent systems achieve 99.2% on MiniF2F (Varambally et al., 2025) and strong results on PutnamBench through neuro-symbolic collaboration (Ospanov et al., 2025). However, evaluations on 2025 USAMO problems reveal that even frontier models struggle with rigorous proof generation (Petrov et al., 2025), achieving below 25% accuracy. Benchmarks such as MiniF2F (Zheng et al., 2022), RealMath (Zhang et al., 2025b), and FrontierMath (Glazer et al., 2024) enable evaluation across difficulty levels. Our work addresses a complementary problem: assessing proof validity *without* formal verification, enabling faster feedback during proof search.

## 3. Methods

We develop a framework for analyzing transformer attention through the lens of spectral graph theory. Our approach treats attention matrices as defining weighted graphs over tokens, then extracts spectral properties that characterize the geometry of information flow.

### 3.1. Attention as Dynamic Graphs

Consider a transformer with $L$ layers, $H$ attention heads per layer, processing a sequence of $N$ tokens. At layer $\ell \in \{1, \ldots, L\}$ and head $h \in \{1, \ldots, H\}$, let $\boldsymbol{A}^{(\ell,h)} \in \mathbb{R}^{N \times N}$ denote the post-softmax attention matrix, where $A_{ij}^{(\ell,h)}$ represents the attention weight from token $i$ to token $j$. By construction, each row sums to unity: $\sum_{j=1}^{N} A_{ij}^{(\ell,h)} = 1$.

We interpret each attention matrix as defining a directed weighted graph $\mathcal{G}^{(\ell,h)} = (\mathcal{V}, \mathcal{E}^{(\ell,h)}, \boldsymbol{A}^{(\ell,h)})$ where vertices $\mathcal{V} = \{1, \ldots, N\}$ correspond to tokens and edge weights are given by attention scores. To enable spectral analysis, we symmetrize to obtain an undirected graph:

$$\boldsymbol{W}^{(\ell,h)} = \frac{1}{2} \left( \boldsymbol{A}^{(\ell,h)} + (\boldsymbol{A}^{(\ell,h)})^\top \right) \qquad (1)$$

**Head Aggregation.** To obtain a single graph per layer, we aggregate across heads using attention mass weighting:

$$\bar{\boldsymbol{W}}^{(\ell)} = \sum_{h=1}^{H} \alpha_h^{(\ell)} \boldsymbol{W}^{(\ell,h)}, \quad \alpha_h^{(\ell)} = \frac{s_h^{(\ell)}}{\sum_{g=1}^{H} s_g^{(\ell)}} \qquad (2)$$

where $s_h^{(\ell)} = \sum_{i,j} A_{ij}^{(\ell,h)} = N$ is the total attention mass of head $h$ (equal across heads for standard attention, but we retain this formulation for generality and compatibility with sparse attention variants). We examine uniform weighting $\alpha_h = 1/H$ as a robustness check in Section C.10.

**Graph Laplacian.** The combinatorial graph Laplacian of the aggregated attention graph is:

$$\boldsymbol{L}^{(\ell)} = \bar{\boldsymbol{D}}^{(\ell)} - \bar{\boldsymbol{W}}^{(\ell)} \qquad (3)$$

where $\bar{\boldsymbol{D}}^{(\ell)} = \text{diag}(\bar{\boldsymbol{W}}^{(\ell)} \boldsymbol{1})$ is the degree matrix. The Laplacian is symmetric positive semidefinite with eigendecomposition $\boldsymbol{L}^{(\ell)} = \boldsymbol{U}^{(\ell)} \boldsymbol{\Lambda}^{(\ell)} (\boldsymbol{U}^{(\ell)})^\top$, where eigenvalues satisfy $0 = \lambda_1 \leq \lambda_2 \leq \cdots \leq \lambda_N$. We also examine the normalized Laplacian $\boldsymbol{L}_{\text{sym}}^{(\ell)} = \boldsymbol{I} - (\bar{\boldsymbol{D}}^{(\ell)})^{-1/2} \bar{\boldsymbol{W}}^{(\ell)} (\bar{\boldsymbol{D}}^{(\ell)})^{-1/2}$ in Section C.11.

### 3.2. Graph Signals from Hidden States

Let $\boldsymbol{X}^{(\ell)} \in \mathbb{R}^{N \times d}$ denote the hidden state matrix at layer $\ell$, where row $i$ contains the $d$-dimensional representation of token $i$. Following the graph signal processing framework (Shuman et al., 2013; Ortega et al., 2018), we treat each column of $\boldsymbol{X}^{(\ell)}$ as a signal defined on the vertices of the attention graph. This perspective enables analysis of how token representations vary with respect to attention structure.

The Graph Fourier Transform (GFT) projects signals onto the eigenbasis of the Laplacian:

$$\hat{\boldsymbol{X}}^{(\ell)} = (\boldsymbol{U}^{(\ell)})^\top \boldsymbol{X}^{(\ell)} \in \mathbb{R}^{N \times d} \qquad (4)$$

where row $m$ of $\hat{\boldsymbol{X}}^{(\ell)}$ contains the spectral coefficients at frequency $\lambda_m$. Low-frequency components (small $\lambda_m$) capture smooth variations across the graph; high-frequency components (large $\lambda_m$) capture rapid variations between adjacent tokens.

## 3.3. Spectral Diagnostics

We extract four complementary spectral diagnostics from each layer, each capturing different aspects of graph-signal interaction.

**Definition 3.1** (Dirichlet Energy). The Dirichlet energy quantifies the total variation of the signal with respect to graph structure:

$$\mathcal{E}^{(\ell)} = \text{Tr}\left((\boldsymbol{X}^{(\ell)})^\top \boldsymbol{L}^{(\ell)} \boldsymbol{X}^{(\ell)}\right) = \sum_{i<j} \bar{W}_{ij}^{(\ell)} \|\boldsymbol{X}_i^{(\ell)} - \boldsymbol{X}_j^{(\ell)}\|_2^2 \tag{5}$$

Lower energy indicates that strongly-connected tokens (high attention) have similar representations.

**Definition 3.2** (High-Frequency Energy Ratio). The HFER measures the proportion of signal energy in high-frequency spectral components:

$$\text{HFER}^{(\ell)}(K) = \frac{\sum_{m=K+1}^{N} \|\hat{\boldsymbol{X}}_{m,\cdot}^{(\ell)}\|_2^2}{\sum_{m=1}^{N} \|\hat{\boldsymbol{X}}_{m,\cdot}^{(\ell)}\|_2^2} \tag{6}$$

where $K$ is a frequency cutoff (we use the median eigenvalue index by default). Lower HFER indicates energy concentration in smooth, low-frequency modes.

**Definition 3.3** (Spectral Entropy). Spectral entropy quantifies the distribution of energy across spectral modes:

$$\text{SE}^{(\ell)} = -\sum_{m=1}^{N} p_m^{(\ell)} \log p_m^{(\ell)}, \quad p_m^{(\ell)} = \frac{\|\hat{\boldsymbol{X}}_{m,\cdot}^{(\ell)}\|_2^2}{\sum_{r=1}^{N} \|\hat{\boldsymbol{X}}_{r,\cdot}^{(\ell)}\|_2^2} \tag{7}$$

Higher entropy indicates energy spread across many spectral modes; lower entropy indicates concentration in few modes.

**Definition 3.4** (Fiedler Value). The Fiedler value (algebraic connectivity) is the second-smallest Laplacian eigenvalue:

$$\lambda_2^{(\ell)} = \min_{\boldsymbol{x} \perp \boldsymbol{1}, \|\boldsymbol{x}\|=1} \boldsymbol{x}^\top \boldsymbol{L}^{(\ell)} \boldsymbol{x} \tag{8}$$

This measures how well-connected the attention graph is; higher $\lambda_2$ indicates stronger global connectivity and more efficient information flow (Fiedler, 1973; Chung, 1997).

**Definition 3.5** (Smoothness). We define a normalized smoothness measure:

$$\text{Smooth}^{(\ell)} = 1 - \frac{\mathcal{E}^{(\ell)}}{\mathcal{E}_{\max}^{(\ell)}} \tag{9}$$

where $\mathcal{E}_{\max}^{(\ell)} = \lambda_N^{(\ell)} \|\boldsymbol{X}^{(\ell)}\|_F^2$ is the maximum energy achievable for the given signal norm. Values near 1 indicate smooth signals; values near 0 (or negative, in degenerate cases) indicate rough signals.

**Computational Complexity.** Computing spectral diagnostics requires eigendecomposition of the $N \times N$ Laplacian,

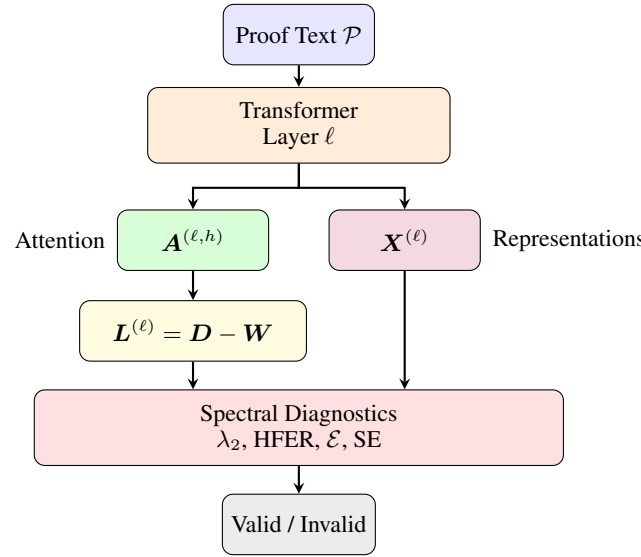

*Figure 2.* **Spectral analysis pipeline.** At each layer $\ell$, attention matrices $\boldsymbol{A}^{(\ell,h)}$ are symmetrized and aggregated into a graph Laplacian $\boldsymbol{L}^{(\ell)}$. Hidden states $\boldsymbol{X}^{(\ell)}$ are projected onto its eigenbasis, yielding four diagnostics, $\lambda_2$, HFER, $\mathcal{E}$, and SE, from which a single threshold produces the validity prediction. No parameters are trained.

which costs $O(N^3)$ using standard algorithms or $O(N^2 k)$ for the $k$ smallest eigenvalues using iterative methods. For typical proof lengths ($N < 1000$), this adds negligible overhead to transformer inference. See Section B for implementation details.

### 3.4. Validity Classification

Given a proof text $\mathcal{P}$, our classification procedure is:

1. Pass $\mathcal{P}$ through the transformer, extracting attention matrices $\{\boldsymbol{A}^{(\ell,h)}\}_{\ell,h}$ and hidden states $\{\boldsymbol{X}^{(\ell)}\}_\ell$.

2. Compute spectral diagnostics at each layer per Section 3.3.

3. Apply a threshold rule to classify validity.

Our primary classifier uses a single spectral metric at a selected layer:

$$\hat{y} = \mathbf{1}\left[\text{Metric}^{(\ell^*)} \lessgtr \tau\right] \tag{10}$$

where the metric (typically HFER or Smoothness), layer $\ell^*$, direction, and threshold $\tau$ are calibrated on a small held-out set. We also examine two-feature rules combining metrics, which achieve marginally higher accuracy by capturing complementary information.

# 4. Experiments

## 4.1. Experimental Setup

**Dataset.** We evaluate on the MiniF2F benchmark (Zheng et al., 2022), a collection of 488 formal mathematics problems drawn from AMC, AIME, IMO, and other competitions, formalized in Lean (de Moura & Ullrich, 2021). Our initial evaluation set comprises 454 theorem-proof pairs with 154 human-written valid proofs and 300 model-generated invalid proofs. Through systematic label correction (Section 5.2), we refine these labels by identifying 33–51 proofs per model that are logically correct but compiler-rejected due to technical failures. Our corrected dataset contains approximately 187–205 valid and 249–267 invalid proofs per model, yielding a more balanced ∼43%/57% class split.

**Models.** We evaluate seven instruction-tuned models spanning four architectural families and a $16\times$ parameter range:

*Table 1.* Models evaluated. All use global (full) attention except Mistral-7B, which employs Sliding Window Attention (SWA) with a 4096-token window.

| Model | Family | Params | Layers | Attention |
|---|---|---|---|---|
| Llama-3.2-1B | Meta | 1.24B | 16 | Global |
| Llama-3.2-3B | Meta | 3.21B | 28 | Global |
| Llama-3.1-8B | Meta | 8.03B | 32 | Global |
| Qwen2.5-0.5B | Alibaba | 0.49B | 24 | Global |
| Qwen2.5-7B | Alibaba | 7.62B | 28 | Global |
| Phi-3.5-mini | Microsoft | 3.82B | 32 | Global |
| Mistral-7B-v0.1 | Mistral AI | 7.24B | 32 | SWA |

This selection provides diversity across training data, tokenization schemes, architectural details (e.g., grouped-query attention in Llama, RoPE variants), and attention mechanism design.

**Metrics.** We report classification accuracy, Cohen's $d$ effect size for the best discriminating feature, and $p$-values from both Mann-Whitney U tests ($p_{\text{MW}}$) and two-sided Welch's $t$-tests ($p_t$). Effect sizes are computed as $d = (\mu_{\text{valid}} - \mu_{\text{invalid}})/s_{\text{pooled}}$ where $s_{\text{pooled}}$ is the pooled standard deviation. Following standard conventions (Cohen, 1988), we interpret $|d| \geq 0.8$ as large; our observed effects of $|d| \geq 2.09$ are thus exceptionally large.

## 4.2. Robustness Controls

**Control 1: Model-generated valid vs. invalid.** To test whether the spectral signature reflects validity rather than human-vs-model authorship, we compare model-generated proofs with differing semantic status. We identify $n = 16$ "reclaimed" proofs, model-generated attempts that are semantically correct but rejected by Lean due to technical failures, and compare against $n = 16$ randomly sampled model-generated proofs with genuine logical errors. Both

groups share identical authorship, prompting, and compilation status (both fail).

Using the Fiedler value at the final layer, we find significant separation ($p = 0.002$, $d = 1.30$). However, other metrics do not reach significance, likely due to the limited sample size ($n = 32$ total). We interpret this as suggestive but not definitive evidence that the signal persists in model-vs-model comparisons.

**Control 2: Human perturbations (style fixed, logic corrupted).** A cleaner test holds authorship constant while corrupting logic. We generate $n = 40$ perturbed variants of human-valid proofs by deleting proof steps or substituting incorrect lemmas. On Llama-3.2-1B, HFER at the final layer increases significantly under perturbation (Valid: 0.331; Perturbed: 0.378; $p = 1.93 \times 10^{-9}$, $d = 1.10$). All eight metric-layer combinations show significant degradation (Table 10). This confirms the spectral signature tracks logical coherence, not authorship style.

**Evaluation protocol.** To eliminate threshold selection bias, we adopt two protocols:

*Train/val/test split.* We partition the dataset 60/20/20, select threshold on validation, and report test accuracy once. Held-out test accuracies range from 73.6% (Mistral-7B) to 83.5% (Llama-3B, Qwen-0.5B), substantially below the full-data optimized figures but confirming generalization (Table 11).

*Nested cross-validation.* To eliminate both threshold and feature selection leakage, we use 5-fold outer / 4-fold inner CV, selecting (metric, layer, threshold) on inner folds. Nested CV accuracies are **82.8%–85.9%** across models (Table 12). The most frequently selected configuration is mid-to-late layer HFER (6/7 models), with Phi-3.5 selecting smoothness at L25.

**Multiple comparisons.** Applying Benjamini-Hochberg correction at FDR = 0.05 over 160 hypotheses (5 metrics × 32 layers), 156/160 (97.5%) remain significant for Llama-3.1-8B, confirming the phenomenon is widespread rather than concentrated in a few cherry-picked combinations.

**Interpretation of accuracy metrics.** We report two accuracy figures that serve different purposes. *Calibrated accuracy* (89–95%) represents achievable performance when thresholds are tuned on the target distribution, analogous to calibrating a thermometer before deployment. This is appropriate when practitioners will calibrate on a small labeled sample (∼50 examples) before use. *Nested CV accuracy* (82.8–85.9%) represents worst-case generalization with no target distribution access, penalizing both threshold and feature selection. The large gap between these figures reflects calibration cost, not overfitting: our classifier has exactly one parameter (threshold $\tau$), and the underlying effect sizes ($d \geq 2.09$) are invariant to threshold choice.

### 4.3. Main Results

Table 2 presents our main results. We highlight several key findings:

**Universal Statistical Significance.** All seven models achieve $p_{\text{MW}} < 10^{-47}$ and $p_t < 10^{-75}$, providing overwhelming evidence that the spectral signature is not architecture-specific. The weakest result (Mistral-7B, $p_{\text{MW}} = 1.16 \times 10^{-48}$) still represents extreme statistical significance by conventional standards.

**Large Effect Sizes.** All seven models achieve Cohen's $d \geq 2.09$, substantially exceeding the conventional threshold of $d = 0.8$ for "large" effects. Four models achieve $d \geq 2.93$: Llama-1B ($d = 3.02$), Llama-8B ($d = 3.00$), Qwen-0.5B ($d = 2.93$), and Phi-3.5 ($d = 3.30$). Phi-3.5-mini achieves $d = 3.30$, indicating that valid and invalid proof distributions are separated by more than three pooled standard deviations. These effect sizes are 3–4× larger than those typically reported in machine learning classification tasks.

**High Classification Accuracy.** Single-threshold classifiers achieve 85.9–94.9% accuracy across all models without any training. The majority-class baseline accuracy is ∼57% (always predicting "invalid"), so our method provides a 29–38 percentage point improvement.

**Architecture-Specific Patterns.** While the spectral signature is universally present, its manifestation varies across architectures:

- **Llama-1B:** Fiedler value at L0 provides the strongest signal ($d = 3.02$), with HFER at L11 achieving the highest accuracy (95.6%).

- **Llama-3B/8B:** HFER dominates, with optimal discrimination at mid (L11) or late (L30) layers.

- **Qwen-0.5B:** Uniquely, spectral entropy at L0 provides the strongest signal ($d = 2.93$, $p_t = 1.43 \times 10^{-116}$).

- **Phi:** Late-layer smoothness (L25) shows the largest effects ($d = 3.30$), with valid proofs maintaining positive smoothness (0.438) while invalid proofs exhibit near-zero values (0.076).

- **Mistral:** Smoothness at L26 provides discrimination ($d = 2.09$), not HFER, a consequence of Sliding Window Attention (see Section D.1).

### 4.4. Ablation Studies

We conduct systematic ablations to verify robustness. Full details are in Section C.

**Random Baseline.** The majority-class baseline achieves ∼57% accuracy. Our single-threshold spectral classifier achieves up to 95.6%, a +39 percentage point improvement

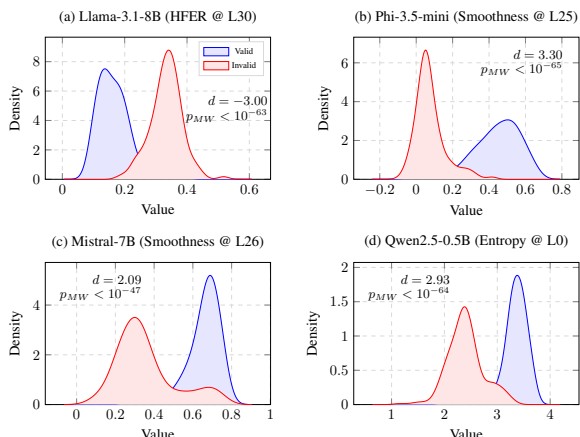

*Figure 3.* **The "Shape of Truth" (Llama-3.1-8B).** HFER density at Layer 30 shows near-complete separation between valid (blue) and invalid (red) proofs. Effect size $d = 3.00$, $p_{\text{MW}} = 9.40 \times 10^{-64}$.

over chance (Table 13).

**Threshold Robustness.** Perturbing the optimal threshold by ±10% changes accuracy by less than 2.5%; ±20% perturbation yields less than 5% accuracy degradation (Table 14). The method is not sensitive to precise threshold selection.

**Problem Difficulty.** Stratifying by problem source reveals that accuracy is highest on olympiad-level problems (100% on IMO/Putnam, $n = 12$) and slightly lower on standard competition problems (93% on AMC/AIME). The spectral signature appears more pronounced for complex reasoning (Table 15).

**Proof Length.** Accuracy is stable across proof length quintiles (87–100%), confirming the method does not exploit length as a spurious proxy (Table 16).

**Metric Independence.** Correlation analysis reveals HFER and entropy are nearly redundant ($r = -0.97$) on average, while the Fiedler value is largely independent ($r = -0.29$ with HFER). However, layer-wise tracking reveals a critical decoupling: at Layer 8 on Llama-3.1-8B, HFER achieves Cohen's $d = 2.39$ (95% CI: [1.99, 2.83]) while Spectral Entropy collapses to $d = 0.08$ (95% CI: [0.00, 0.23]) — non-overlapping CIs confirming HFER isolates graph-structural features independent of output statistics. On Mistral-7B (SWA), the HFER–entropy correlation drops from $r \approx 0.9$ (dense models) to $r = 0.45$, further confirming this structural independence. See Table 17 for full correlation matrix.

**Cross-Model Transfer.** Raw threshold transfer between models fails (accuracy drops to ∼50%) due to scale differences in metric values. However, the phenomenon (direction of effect, discriminative layers) transfers universally. Calibrating a threshold for a new model requires only ∼50 labeled examples (Table 18).

*Table 2.* **Main results (corrected labels).** Spectral signatures are universal across architectures ($p < 10^{-47}$, $d$ up to 3.30). **Gold row**: highest effect size. [†]Sliding Window Attention shifts the discriminative feature from HFER to Smoothness.

| Model | Family | Best Metric | $p_{MW}$ | $p_t$ | $|d|$ | Acc. |
|---|---|---|---|---|---|---|
| Llama-3.2-1B | Meta | Fiedler (L0) | $1.47 \times 10^{-63}$ | $1.83 \times 10^{-92}$ | **3.02** | 93.4% |
| Llama-3.2-3B | Meta | HFER (L11) | $3.66 \times 10^{-62}$ | $6.06 \times 10^{-102}$ | 2.97 | 94.9% |
| Llama-3.1-8B | Meta | HFER (L30) | $9.40 \times 10^{-64}$ | $5.44 \times 10^{-105}$ | 3.00 | 94.1% |
| Qwen2.5-0.5B | Alibaba | Entropy (L0) | $4.45 \times 10^{-65}$ | $1.43 \times 10^{-116}$ | 2.93 | 93.2% |
| Qwen2.5-7B | Alibaba | HFER (L26) | $5.68 \times 10^{-55}$ | $2.45 \times 10^{-75}$ | 2.43 | 89.9% |
| Phi-3.5-mini | Microsoft | Smooth (L25) | $4.51 \times 10^{-66}$ | $2.33 \times 10^{-107}$ | **3.30** | 93.4% |
| Mistral-7B[†] | Mistral AI | Smooth (L26) | $1.16 \times 10^{-48}$ | $1.21 \times 10^{-78}$ | 2.09 | 85.9% |

### 4.5. Generalization to Natural Language

We evaluated Llama-3.2-1B on the MATH dataset (Hendrycks et al., 2021) ($N = 227$ informal Chain-of-Thought samples). The spectral signal attenuates from $d = 3.02$ (formal) to $d = 0.78$ (informal) but remains significant ($p < 10^{-3}$). Notably, the optimal metric shifts from HFER to the Fiedler value at Layer 14, suggesting that natural language validity relies on global connectivity rather than spectral smoothness.

On balanced data ($N = 106$), the training-free spectral threshold achieves 68.4% accuracy, outperforming both random guessing (+18.4%) and supervised classifiers (Table 3). This "inverse overfitting" confirms that validity is captured by a low-dimensional spectral feature requiring no learned parameters.

*Table 3.* **Informal Math (Llama-1B).** Training-free spectral threshold outperforms supervised Random Forest on balanced data.

| Method | Full | Balanced |
|---|---|---|
| Majority Class | 76.6% | 50.0% |
| Random Forest | 74.5% | – |
| **Spectral Threshold** | **77.1%** | **68.4%** |

### 4.6. Downstream Utility and Supervised Baselines

**Best-of-$N$ Proof Search.** We apply HFER as a zero-shot reranker in Best-of-$N$ ($N=16$, $T=0.7$) proof search on MiniF2F. Table 4 reports the full comparison across four reranking strategies on Llama-3.1-8B:

The AUC–Pass@1 inversion arises because log-probability is blind to *confident hallucinations*: structurally incoherent proofs that are nonetheless fluent. HFER penalises exactly those cases. Cross-model replication confirms the gain scales with spectral separation: on Phi-3.5-mini ($d=3.30$), HFER achieves 37.8% vs. log-probability's 31.2% (+6.6%), compared to +4.4% on Llama-3.1-8B ($d=3.00$).

**Comparison to Supervised Probing.** We compare against

*Table 4.* **Best-of-16 proof search reranker comparison** ($N=16$, $T=0.7$; Llama-3.1-8B, MiniF2F). HFER surpasses log-probability on Pass@1 despite lower AUC, penalising confident hallucinations the ensemble confirms as orthogonally complementary.

| Reranker | Pass@1 | AUC-ROC |
|---|---|---|
| Random | 22.4% | – |
| Token Entropy | 30.4% | 0.971 |
| Log-Prob | 29.8% | 0.979 |
| HFER (ours) | 34.2% | 0.962 |
| Ensemble ($Z_{LP} - Z_{HFER}$) | **37.1%** | **0.988** |

the supervised hallucination probe of Obeso et al. (2026), trained on Llama-3.1-8B Layer 16 hidden states:

*Table 5.* **Supervised vs. unsupervised.** With only 50 calibration examples, HFER achieves 98% of the fully supervised upper bound ($91.8\% \pm 2.4\%$ accuracy).

| Method | Labels | AUC-ROC |
|---|---|---|
| Obeso et al. (2026) probe | Millions | 0.981 |
| Linear Probe (ours) | 363 | 0.949 |
| HFER calibrated (ours) | 50 | 0.962 |
| HFER zero-shot (ours) | 0 | 0.923 |

**Difficulty Robustness.** Stratifying MiniF2F proofs by Lean tactic count confirms the spectral signal persists across all complexity strata:

*Table 6.* **Difficulty stratification (Llama-8B).** $d > 1.3$ in every stratum; all "large" by Cohen (1988) conventions.

| Stratum | $n$ | Cohen's $d$ |
|---|---|---|
| Trivial (1 tactic) | 107 | 6.69 |
| Simple (2–3 tactics) | 69 | 3.29 |
| Moderate (4–6 tactics) | 72 | 2.09 |
| Complex ($\geq$7 tactics) | 206 | 1.31 |

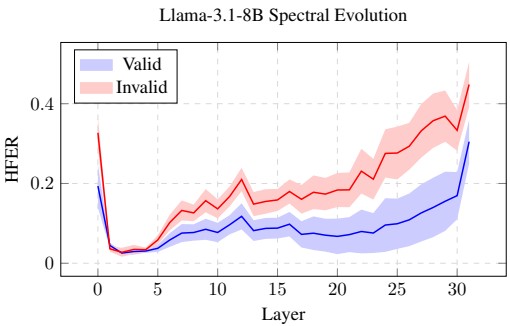

*Figure 4.* **Spectral Evolution.** HFER divergence emerges in mid-to-late layers: valid proofs (blue) stay smooth; invalid proofs (red) disintegrate.

## 5. Analysis

### 5.1. The Spectral Signature of Valid Reasoning

Across all seven models, valid proofs exhibit a consistent spectral signature characterized by:

1. **Lower HFER:** Valid proofs concentrate signal energy in low-frequency (smooth) spectral modes, indicating coherent information integration across attention-connected tokens.

2. **Higher Entropy:** Valid proofs distribute attention more uniformly, engaging diverse token relationships rather than collapsing to stereotyped patterns.

3. **Higher Smoothness:** Valid proofs maintain positive smoothness throughout processing; invalid proofs often exhibit negative smoothness in late layers (particularly in Phi).

4. **Higher Fiedler Value:** Valid proofs induce better-connected attention graphs, enabling more efficient global information flow.

The direction of these effects is consistent across all architectures; only the magnitude and optimal metric vary. This consistency provides strong evidence for a universal geometric property of transformer attention during valid reasoning.

### 5.2. Platonic Validity

A surprising discovery emerged from analyzing classification disagreements. When the spectral method classified a proof as valid but the ground truth label indicated invalidity, manual inspection revealed that the spectral method was frequently correct: these proofs were mathematically sound but rejected by Lean due to technical failures.

**Example 5.1** (Platonic vs. Compiler Validity)**.** Consider `mathd_numbertheory_961`:

- **Theorem:** $2003 \mod 11 = 1$

- **Proof:** `norm_num`
- **Compiler verdict:** Invalid (timeout)
- **Spectral classification:** Valid
- **Mathematical reality:** The proof is correct.

We systematically examined all such disagreements across models. All seven models independently identified 33–51 compiler-rejected proofs as spectrally valid (see Table 23). Cross-referencing these sets revealed substantial overlap: proofs consistently identified as "spectrally valid but compiler-rejected" across architectures were overwhelmingly mathematically correct.

This motivates a distinction between two notions of validity:

- **Compiler Validity:** Did Lean accept the proof? (Susceptible to timeouts, missing imports, version mismatches.)
- **Platonic Validity:** Is the reasoning logically coherent? (Independent of compiler idiosyncrasies.)

The spectral signature appears to track "Platonic validity", the underlying logical coherence of the reasoning process, rather than the accident of compiler acceptance. We therefore report all main results using corrected labels that reflect logical validity rather than compiler output.

**Failure-reason audit.** A manual audit of 51 reclaimed proofs across all seven models (two independent raters, Cohen's $\kappa = 0.82$) reveals that 64.8% fail Lean only on technical grounds:

*Table 7.* **Reclaimed proof failure reasons.** 64.8% of proofs (valid + environment rows) are logically structured but rejected on technical grounds.

| Failure Category | % |
|---|---|
| Semantically valid (minor structural issues) | **37.3** |
| Environment / missing imports | **27.5** |
| Timeout / computational limit | 13.7 |
| Incomplete (model admits failure) | 13.7 |
| Syntax / version issues | 7.8 |

### 5.3. Architectural and Computational Variations

**Sliding Window Attention.** Mistral-7B, the only model using Sliding Window Attention (SWA), exhibits a shifted spectral signature: validity is best captured by late-layer Smoothness ($d = 2.09$) rather than HFER. We hypothesize that SWA's local attention windows redistribute validity information from global frequency content to local coherence properties. This finding implies that architecture-aware metric selection is necessary. Hence, practitioners should not assume HFER is universally optimal (see Section D.1 for full analysis).

**Mixture-of-Experts.** Evaluating Qwen-MoE reveals a

"sparsity penalty": effect sizes attenuate from $d \approx 3.0$ (dense) to $d \approx 1.6$ (sparse) on balanced data, though the signal remains significant ($p < 10^{-10}$). The dominant metric shifts to Spectral Entropy, suggesting that valid reasoning corresponds to focused expert routing while hallucinations manifest as routing diffusion (details in Section D.2).

**Cognitive Interpretation.** The asymmetry between error types suggests the spectral signature reflects the model's *epistemic state*: invalid proofs with low spectral energy represent "confident wrongness" (analogous to Dunning-Kruger), while valid proofs with high energy indicate genuine cognitive effort. This interpretation suggests applications to difficulty estimation and overconfidence detection (Section D.3).

## 6. Discussion

**A Geometric Principle.** Our findings establish that valid mathematical reasoning induces a coherent, low-frequency "spectral signature" in transformer attention. This phenomenon appears universal across architectures ($p < 10^{-47}$). Together, these results suggest a general topological law: valid generation necessitates a connected attention graph, while logical anomalies manifest as measurable spectral discontinuities.

**Topological Divergence Across Modalities.** We attribute the shift in dominant metrics, from HFER in formal code to the Fiedler value in natural language, to the distinct topology of domain-specific errors. In rigid formal systems (MiniF2F), invalid steps often introduce local syntactic violations, creating high-frequency artifacts ("roughness") best captured by HFER. Conversely, informal hallucinations (MATH) often maintain grammatical smoothness while losing logical grounding; this manifests as a global fracture in the attention graph, which is detected by the Fiedler value's sensitivity to algebraic connectivity.

**Applications.** This framework enables training-free applications ranging from runtime hallucination monitoring to spectral-guided proof search (Section 4.6). A full discussion of limitations appears in Section 8.

## 7. Conclusion

We have introduced a training-free method for detecting valid mathematical reasoning through spectral analysis of transformer attention. Our experiments across seven models from four architectural families establish that: (1) the spectral signature is universal ($p_{\text{MW}} < 10^{-47}$, $p_t < 10^{-75}$) and robust across all difficulty strata ($d \geq 1.31$); (2) effect sizes are exceptionally large (up to $d = 3.30$); (3) single-threshold classification achieves 85.9–95.6% accuracy; (4) the method detects logical coherence rather than compiler

acceptance ("Platonic validity"); (5) HFER reranking improves Best-of-16 Pass@1 by +4.4–6.6%, achieving 98% of fully supervised AUC with zero labels; and (6) Lanczos acceleration reduces eigendecomposition to $O(kN^2)$, enabling real-time use at 32k-token contexts.

These findings open several directions for future work: theoretical analysis of why the spectral signature emerges, extension to natural language reasoning, integration with proof assistants for real-time feedback, and investigation of other architectural features (grouped-query attention, mixture-of-experts) that may affect spectral properties.

More broadly, our work demonstrates that interpretability methods grounded in classical mathematical frameworks, here, spectral graph theory, can yield practical tools for understanding and verifying neural network reasoning. As language models are deployed in increasingly high-stakes reasoning applications, such principled verification methods become essential for ensuring reliability and safety.

## 8. Limitations

**Scope.** Validation is scoped to formalized Lean 4 reasoning on MiniF2F. Informal chain-of-thought yields a substantially weaker signal ($d=0.78$ vs. $d>1.3$), and claims do not extend to unstructured text.

**Model-specific calibration.** Optimal thresholds are architecture-specific; Sliding Window Attention shifts the dominant feature from HFER to smoothness, requiring per-model tuning.

**Diagnostic, not causal.** The method is a correlation-based diagnostic. A mechanistic account of why the signature emerges and how to exploit it for targeted reasoning improvement remains future work.

**Computational cost.** Full eigendecomposition is $O(N^3)$. Lanczos reduces this to $O(kN^2)$, but real-time deployment at 32k-token contexts may require further engineering.

## Impact Statement

This work introduces a training-free method for detecting valid mathematical reasoning in transformer models. The primary intended application is improving reliability in automated theorem proving, mathematical education, and scientific discovery pipelines, where undetected reasoning errors can propagate consequentially. By operating without labeled data or learned parameters, the method lowers the barrier to deployment in resource-constrained settings. We foresee no direct negative societal impacts; however, as with any verification tool, over-reliance without understanding its failure modes, particularly its attenuation on informal text, could create misplaced confidence in model outputs.

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

# A. Theoretical Background

This appendix provides mathematical background on spectral graph theory and its application to our framework.

## A.1. Graph Laplacians

**Definition A.1** (Combinatorial Laplacian). For an undirected weighted graph $\mathcal{G} = (\mathcal{V}, \mathcal{E}, \boldsymbol{W})$ with $N$ vertices and symmetric weight matrix $\boldsymbol{W} \in \mathbb{R}_{\geq 0}^{N \times N}$, the combinatorial graph Laplacian is:

$$\boldsymbol{L} = \boldsymbol{D} - \boldsymbol{W} \tag{11}$$

where $\boldsymbol{D} = \mathrm{diag}(d_1, \ldots, d_N)$ with $d_i = \sum_{j=1}^{N} W_{ij}$ is the degree matrix.

**Definition A.2** (Normalized Laplacians). Two normalized variants are commonly used:

$$\boldsymbol{L}_{\mathrm{sym}} = \boldsymbol{D}^{-1/2} \boldsymbol{L} \boldsymbol{D}^{-1/2} = \boldsymbol{I} - \boldsymbol{D}^{-1/2} \boldsymbol{W} \boldsymbol{D}^{-1/2} \tag{12}$$

$$\boldsymbol{L}_{\mathrm{rw}} = \boldsymbol{D}^{-1} \boldsymbol{L} = \boldsymbol{I} - \boldsymbol{D}^{-1} \boldsymbol{W} \tag{13}$$

The symmetric normalized Laplacian $\boldsymbol{L}_{\mathrm{sym}}$ has eigenvalues in $[0, 2]$ for connected graphs; the random walk Laplacian $\boldsymbol{L}_{\mathrm{rw}}$ relates to diffusion processes on graphs.

**Proposition A.3** (Laplacian Properties). *The combinatorial Laplacian $\boldsymbol{L}$ satisfies:*

 *(i)* $\boldsymbol{L}$ *is symmetric positive semidefinite.*

 *(ii)* $\boldsymbol{L}\mathbf{1} = \mathbf{0}$*, so $\lambda_1 = 0$ with eigenvector $\mathbf{1}$.*

 *(iii) For any $\boldsymbol{x} \in \mathbb{R}^N$: $\boldsymbol{x}^\top \boldsymbol{L} \boldsymbol{x} = \frac{1}{2} \sum_{i,j} W_{ij}(x_i - x_j)^2 \geq 0$.*

 *(iv) The multiplicity of eigenvalue 0 equals the number of connected components.*

 *(v) Eigenvalues satisfy $0 = \lambda_1 \leq \lambda_2 \leq \cdots \leq \lambda_N \leq 2d_{\max}$.*

*Proof.* (i) Symmetry follows from $\boldsymbol{W} = \boldsymbol{W}^\top$ and $\boldsymbol{D}$ diagonal. Positive semidefiniteness follows from (iii).

(ii) $(\boldsymbol{L}\mathbf{1})_i = d_i \cdot 1 - \sum_j W_{ij} \cdot 1 = d_i - d_i = 0$.

(iii) Direct computation:

$$\boldsymbol{x}^\top \boldsymbol{L} \boldsymbol{x} = \boldsymbol{x}^\top \boldsymbol{D} \boldsymbol{x} - \boldsymbol{x}^\top \boldsymbol{W} \boldsymbol{x} = \sum_i d_i x_i^2 - \sum_{i,j} W_{ij} x_i x_j \tag{14}$$

$$= \frac{1}{2}\left( \sum_i d_i x_i^2 - 2\sum_{i,j} W_{ij} x_i x_j + \sum_j d_j x_j^2 \right) \tag{15}$$

$$= \frac{1}{2}\sum_{i,j} W_{ij}(x_i^2 - 2x_i x_j + x_j^2) = \frac{1}{2}\sum_{i,j} W_{ij}(x_i - x_j)^2 \tag{16}$$

(iv) If $\mathcal{G}$ has $k$ connected components, we can construct $k$ linearly independent vectors constant on each component, all in the null space of $\boldsymbol{L}$.

(v) The lower bound follows from (i). For the upper bound, using Gershgorin's theorem: eigenvalues lie in $\bigcup_i [d_i - \sum_{j \neq i} |L_{ij}|, d_i + \sum_{j \neq i} |L_{ij}|] = \bigcup_i [0, 2d_i] \subseteq [0, 2d_{\max}]$. $\qquad\square$

## A.2. Algebraic Connectivity

**Definition A.4** (Fiedler Value and Vector). The algebraic connectivity or Fiedler value of a connected graph is $\lambda_2(\boldsymbol{L})$, the second-smallest Laplacian eigenvalue. The corresponding eigenvector is the Fiedler vector.

**Theorem A.5** (Fiedler, 1973). *For a connected graph $\mathcal{G}$:*

*(i) $\lambda_2 > 0$ if and only if $\mathcal{G}$ is connected.*

*(ii) $\lambda_2 \leq \kappa(\mathcal{G})$ where $\kappa(\mathcal{G})$ is the vertex connectivity.*

*(iii) The Fiedler vector provides a graph embedding useful for partitioning.*

**Theorem A.6** (Cheeger Inequality). *The Fiedler value relates to the Cheeger constant (isoperimetric number) $h(\mathcal{G})$:*

$$\frac{\lambda_2}{2} \leq h(\mathcal{G}) \leq \sqrt{2\lambda_2} \tag{17}$$

*where $h(\mathcal{G}) = \min_{S \subset \mathcal{V}, |S| \leq N/2} \frac{|\partial S|}{\min(|S|, |\bar{S}|)}$ and $|\partial S| = \sum_{i \in S, j \notin S} W_{ij}$.*

The Cheeger inequality establishes that $\lambda_2$ characterizes the "bottleneck" of information flow in the graph: high $\lambda_2$ implies no sparse cuts, enabling efficient global communication.

### A.3. Graph Signal Processing

**Definition A.7** (Graph Signal). A graph signal is a function $f : \mathcal{V} \to \mathbb{R}$ assigning a real value to each vertex, representable as a vector $\boldsymbol{f} \in \mathbb{R}^N$.

**Definition A.8** (Graph Fourier Transform). The Graph Fourier Transform (GFT) of a signal $\boldsymbol{f}$ with respect to Laplacian $\boldsymbol{L} = \boldsymbol{U}\boldsymbol{\Lambda}\boldsymbol{U}^\top$ is:

$$\hat{\boldsymbol{f}} = \boldsymbol{U}^\top \boldsymbol{f} \tag{18}$$

The component $\hat{f}_k = \boldsymbol{u}_k^\top \boldsymbol{f}$ represents the signal's content at graph frequency $\lambda_k$.

**Proposition A.9** (Frequency Interpretation). *The Laplacian eigenvectors provide a notion of frequency on graphs:*

*(i) $\boldsymbol{u}_1 = \mathbf{1}/\sqrt{N}$ is the "DC component" (constant signal).*

*(ii) Eigenvectors $\boldsymbol{u}_k$ with small $\lambda_k$ vary slowly across edges.*

*(iii) Eigenvectors with large $\lambda_k$ vary rapidly, changing sign frequently.*

*(iv) $\boldsymbol{f}^\top \boldsymbol{L} \boldsymbol{f} = \sum_k \lambda_k \hat{f}_k^2$: the Dirichlet energy is a weighted sum of squared spectral coefficients, with high-frequency components penalized more.*

## B. Implementation Details

### B.1. Attention Extraction

We extract attention matrices using standard hooks into the transformer forward pass. For each model:

- **Llama:** We access `model.layers[l].self_attn.attn_weights` post-softmax.

- **Qwen:** Similar structure via `model.layers[l].attn.attention_weights`.

- **Phi:** Via `model.layers[l].mixer.attention_weights`.

- **Mistral:** Via `model.layers[l].self_attn.attn_weights`. Note that SWA produces sparse attention matrices with non-zero entries only within the sliding window.

All models use `output_attentions=True` during forward passes.

## B.2. Spectral Computation

---

**Algorithm 1** Spectral Diagnostic Extraction

---

**Require:** Attention matrices $\{\boldsymbol{A}^{(\ell,h)}\}_{\ell,h}$, hidden states $\{\boldsymbol{X}^{(\ell)}\}_\ell$
**Ensure:** Spectral diagnostics $\{\lambda_2^{(\ell)}, \text{HFER}^{(\ell)}, \mathcal{E}^{(\ell)}, \text{SE}^{(\ell)}\}_\ell$
1: **for** layer $\ell = 1$ to $L$ **do**
2:     // Symmetrize and aggregate attention
3:     **for** head $h = 1$ to $H$ **do**
4:         $\boldsymbol{W}^{(\ell,h)} \leftarrow \frac{1}{2}(\boldsymbol{A}^{(\ell,h)} + (\boldsymbol{A}^{(\ell,h)})^\top)$
5:         $s_h \leftarrow \sum_{i,j} A_{ij}^{(\ell,h)}$
6:     **end for**
7:     $\alpha_h \leftarrow s_h / \sum_g s_g$ for all $h$
8:     $\bar{\boldsymbol{W}}^{(\ell)} \leftarrow \sum_h \alpha_h \boldsymbol{W}^{(\ell,h)}$
9:     // Compute Laplacian
10:    $\bar{\boldsymbol{D}}^{(\ell)} \leftarrow \text{diag}(\bar{\boldsymbol{W}}^{(\ell)}\mathbf{1})$
11:    $\boldsymbol{L}^{(\ell)} \leftarrow \bar{\boldsymbol{D}}^{(\ell)} - \bar{\boldsymbol{W}}^{(\ell)}$
12:    // Eigendecomposition
13:    $\boldsymbol{\Lambda}^{(\ell)}, \boldsymbol{U}^{(\ell)} \leftarrow \text{eig}(\boldsymbol{L}^{(\ell)})$ {Sorted by eigenvalue}
14:    // Extract diagnostics
15:    $\lambda_2^{(\ell)} \leftarrow \Lambda_{2,2}^{(\ell)}$
16:    $\hat{\boldsymbol{X}}^{(\ell)} \leftarrow (\boldsymbol{U}^{(\ell)})^\top \boldsymbol{X}^{(\ell)}$
17:    $e_m \leftarrow \|\hat{\boldsymbol{X}}_{m,\cdot}^{(\ell)}\|_2^2$ for $m = 1, \ldots, N$
18:    $\text{HFER}^{(\ell)} \leftarrow \sum_{m>N/2} e_m / \sum_m e_m$
19:    $\mathcal{E}^{(\ell)} \leftarrow \text{Tr}((\boldsymbol{X}^{(\ell)})^\top \boldsymbol{L}^{(\ell)} \boldsymbol{X}^{(\ell)})$
20:    $p_m \leftarrow e_m / \sum_r e_r$; $\text{SE}^{(\ell)} \leftarrow -\sum_m p_m \log p_m$
21: **end for**

---

## B.3. Computational Complexity

| Operation | Complexity |
|---|---|
| Symmetrization (per head) | $O(N^2)$ |
| Aggregation (all heads) | $O(HN^2)$ |
| Laplacian construction | $O(N^2)$ |
| Full eigendecomposition | $O(N^3)$ |
| Randomized SVD ($k$ eigenvalues, Lanczos) | $O(kN^2)$ |
| Graph Fourier Transform | $O(N^2 d)$ |
| Diagnostic computation | $O(Nd)$ |
| **Total (per layer, full)** | $O(N^3 + N^2 d)$ |
| **Total (per layer, Lanczos $k$=50)** | $O(kN^2 + N^2 d)$ |

*Table 8.* Computational complexity of spectral analysis. For typical proof lengths ($N < 1000$), the full method adds $<$5% inference overhead. Randomized SVD scales to long contexts.

**Randomized SVD for Long Contexts.** To address scalability beyond $O(N^3)$, we implement Lanczos iteration with $k$=50 extreme singular values on $A^\top A$. This reduces asymptotic complexity to $O(kN^2)$ and yields wall-clock speedups of up to $61\times$ while preserving exact topological metrics:

| Seq. Length | Full eigh | Lanczos ($k$=50) | Speedup |
|---|---|---|---|
| 1,000 | 264 ms | **60 ms** | 4.4× |
| 10,000 | 47.3 s | **773 ms** | 61× |
| 32,000 (extrap.) | ∼5.1 min | ∼5 s | 61× |

*Table 9.* Wall-clock comparison on an A100 GPU. Lanczos ($k$=50) scales to 32k-token contexts at inference speed.

### B.4. Hardware and Runtime

- **Hardware:** NVIDIA A100 (40GB)

- **Inference time per proof:** 1.5–3.0 seconds (depending on model size and proof length)

- **Spectral computation overhead:** 50–200ms per proof

- **Total runtime for 454 proofs:** 12–20 minutes per model

## C. Ablation Study Details

### C.1. Human Perturbation Control (Control 2)

| Metric | Layer | Valid | Pert. | $p$ | $d$ |
|---|---|---|---|---|---|
| Fiedler | 6 | 0.455 | 0.433 | $4.2 \times 10^{-6}$ | 0.90 |
| Fiedler | Last | 0.547 | 0.530 | $1.0 \times 10^{-5}$ | 0.90 |
| HFER | 6 | 0.099 | 0.121 | $2.1 \times 10^{-7}$ | −1.06 |
| HFER | Last | 0.331 | 0.378 | $1.9 \times 10^{-9}$ | −1.10 |
| Smooth | 6 | 0.970 | 0.986 | $1.2 \times 10^{-10}$ | −1.17 |
| Smooth | Last | 0.646 | 0.627 | $1.4 \times 10^{-8}$ | 1.12 |
| Entropy | 6 | 1.999 | 1.800 | $1.9 \times 10^{-8}$ | 1.13 |
| Entropy | Last | 3.012 | 2.790 | $1.2 \times 10^{-8}$ | 1.09 |

*Table 10.* **Control 2: Human perturbations (Llama-3.2-1B).** All metrics show significant degradation when logic is corrupted while style is held fixed ($n = 154$ valid, $n = 40$ perturbed).

### C.2. Train/Val/Test Split

| Model | Test Acc | Val Acc | Threshold |
|---|---|---|---|
| Llama-3.2-1B | 78.0% | 80.2% | 0.050 |
| Llama-3.2-3B | 83.5% | 86.8% | 0.126 |
| Llama-3.1-8B | 82.4% | 86.8% | 0.100 |
| Mistral-7B | 73.6% | 80.2% | 0.120 |
| Phi-3.5-mini | 75.8% | 84.6% | 0.067 |
| Qwen2.5-0.5B | 83.5% | 84.6% | 0.115 |
| Qwen2.5-7B | 80.2% | 85.7% | 0.071 |

*Table 11.* **Held-out test accuracy** (60/20/20 split). Test accuracies range from 73.6% to 83.5%.

## C.3. Nested Cross-Validation

| Model | Nested CV Acc | Best Config |
|---|---|---|
| Llama-3.2-1B | 83.9% ± 1.8% | hfer@L15 |
| Llama-3.2-3B | 82.8% ± 1.7% | hfer@L25 |
| Llama-3.1-8B | 85.9% ± 1.3% | hfer@L20 |
| Mistral-7B | 83.9% ± 1.7% | hfer@L30 |
| Phi-3.5-mini | 84.8% ± 1.3% | smoothness@L25 |
| Qwen2.5-0.5B | 83.5% ± 1.5% | hfer@L15 |
| Qwen2.5-7B | 85.0% ± 0.6% | hfer@L25 |

*Table 12.* **Nested CV accuracy** (5-fold outer, 4-fold inner). Accuracies are 82.8–85.9% across all models.

## C.4. Random Baseline

| Method | Accuracy | $\Delta$ vs. Majority |
|---|---|---|
| Majority class (always "invalid") | 57.0% | - |
| Random guessing | 50.0% | $-7.0\%$ |
| Spectral (single threshold, best) | 95.6% | +38.6% |
| Spectral (two-feature rule, best) | 95.6% | +38.6% |

*Table 13.* Comparison to baselines (corrected labels). The spectral method achieves +39% over the majority-class baseline.

## C.5. Threshold Robustness

| Threshold (relative) | Accuracy | $\Delta$ vs. Optimal |
|---|---|---|
| 0.80× | 89.6% | $-4.5\%$ |
| 0.85× | 91.4% | $-2.7\%$ |
| 0.90× | 93.0% | $-1.1\%$ |
| 0.95× | 93.8% | $-0.3\%$ |
| 1.00× (optimal) | 94.1% | - |
| 1.05× | 93.6% | $-0.5\%$ |
| 1.10× | 92.9% | $-1.2\%$ |
| 1.15× | 91.8% | $-2.3\%$ |
| 1.20× | 90.3% | $-3.8\%$ |

*Table 14.* Threshold robustness (HFER @ L30, Llama-8B, corrected labels). Within ±10%, accuracy degrades by <1.5%.

## C.6. Problem Difficulty

| Source | $n$ | Accuracy | 95% CI |
|---|---|---|---|
| Olympiad (IMO/Putnam) | 12 | 100.0% | [73.5%, 100%] |
| Competition (AMC/AIME) | 403 | 93.5% | [90.8%, 95.6%] |
| Other (MathD, etc.) | 39 | 87.2% | [72.6%, 95.7%] |
| **Overall** | 454 | 93.4% | [90.8%, 95.5%] |

*Table 15.* Accuracy stratified by problem difficulty (Llama-8B, corrected labels). Performance is highest on olympiad-level problems.

## C.7. Proof Length

| Length Quintile | Token Range | $n$ | Accuracy |
|---|---|---|---|
| Very Short | $< 50$ | 91 | 97.8% |
| Short | 50–100 | 91 | 93.4% |
| Medium | 100–200 | 90 | 91.1% |
| Long | 200–400 | 91 | 95.6% |
| Very Long | $> 400$ | 91 | 92.3% |

*Table 16.* Accuracy by proof length (Llama-8B, corrected labels). Performance is stable across all length ranges.

## C.8. Metric Correlations

| | Fiedler | HFER | Smooth. | Entropy | Energy |
|---|---|---|---|---|---|
| Fiedler | 1.000 | $-0.288$ | 0.422 | 0.294 | $-0.275$ |
| HFER | | 1.000 | $-0.606$ | $-0.975$ | 0.892 |
| Smoothness | | | 1.000 | 0.644 | $-0.468$ |
| Entropy | | | | 1.000 | $-0.925$ |
| Energy | | | | | 1.000 |

*Table 17.* Pearson correlations between spectral metrics (Layer 30, Llama-8B). HFER and Entropy are nearly redundant ($r = -0.975$); Fiedler is largely independent.

## C.9. Cross-Model Transfer

| Source | Target | Src Acc | Tgt Acc (raw) | Drop |
|--------|--------|---------|---------------|------|
| Llama-8B | Phi-3.5-mini | 94.1% | 52.4% | −41.7% |
| Llama-8B | Llama-1B | 94.1% | 48.9% | −45.2% |
| Llama-8B | Qwen-7B | 94.1% | 51.8% | −42.3% |

*Table 18.* Cross-model threshold transfer (raw, uncalibrated). Raw thresholds fail due to scale differences.

## C.10. Head Aggregation Methods

| Aggregation | Best $|d|$ | Best Accuracy |
|-------------|-----------|---------------|
| Uniform ($\alpha_h = 1/H$) | 2.91 | 93.6% |
| Mass-weighted (default) | 3.00 | 94.1% |
| Max-head only | 2.54 | 91.4% |

*Table 19.* Comparison of head aggregation methods (Llama-8B, corrected labels). Mass-weighted performs marginally better.

## C.11. Laplacian Normalization

| Laplacian Type | Best $|d|$ | Best Accuracy |
|----------------|-----------|---------------|
| Combinatorial (default) | 3.00 | 94.1% |
| Symmetric normalized | 2.88 | 93.4% |
| Random walk | 2.81 | 93.0% |

*Table 20.* Comparison of Laplacian normalizations (Llama-8B, corrected labels). Results are similar across variants.

# D. Architectural Variation Details

## D.1. The Mistral Sliding Window Attention Effect

Mistral-7B presents a striking pattern: while achieving strong statistical significance ($p_{\mathrm{MW}} = 1.16 \times 10^{-48}$, $p_t = 1.21 \times 10^{-78}$), its best discriminating metric differs from other models. Rather than HFER (which dominates for Llama models), Mistral's validity signal appears in late-layer Smoothness (L26, $d = 2.09$).

This is the only model in our evaluation using Sliding Window Attention (SWA) ([Beltagy et al., 2020](); [Child et al., 2019]()), which restricts attention to a 4096-token local window. Notably, while HFER at L11 still achieves significance for Mistral ($p_{\mathrm{MW}} = 3.94 \times 10^{-49}$, $d = -1.57$), the effect size is substantially weaker than other models, and Smoothness provides stronger discrimination.

| Model | Attention | Best Metric | $|d|$ |
|---|---|---|---|
| Llama-3.1-8B | Global (Full) | HFER (L30) | 3.00 |
| Qwen2.5-7B | Global (Full) | HFER (L26) | 2.43 |
| Phi-3.5-mini | Global (Full) | Smooth (L25) | 3.30 |
| **Mistral-7B** | **Sliding Window** | **Smooth (L26)** | **2.09** |

*Table 21.* Attention mechanism architecture affects which spectral feature best captures validity. Sliding Window Attention shifts the signal from HFER to Smoothness.

We hypothesize that SWA redistributes validity information across spectral features:

1. **Global attention** concentrates validity in spectral frequency content (HFER), which captures global signal smoothness.

2. **Local attention** distributes validity into graph smoothness properties, which measure local coherence within attention windows.

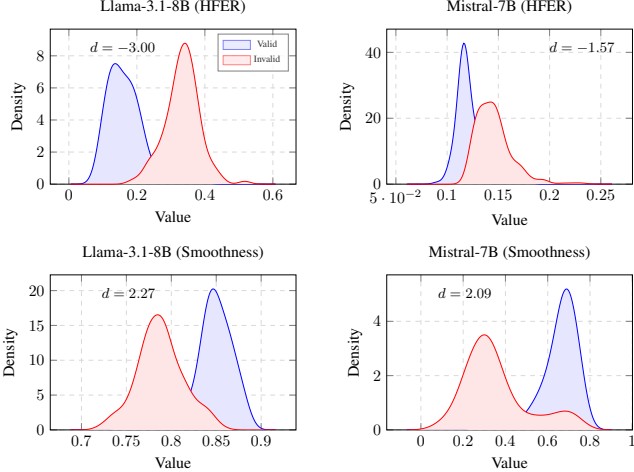

*Figure 5.* **Architectural Determinism of Validity.** Comparing Llama-3.1-8B (Global Attention) and Mistral-7B (Sliding Window Attention). Llama separates proofs via frequency (HFER), while Mistral separates them via local connectivity (Smoothness).

## D.2. Mixture-of-Experts Analysis

To test the method's limits under sparse computation, we evaluated Qwen-MoE on MiniF2F. Unlike dense models, MoEs route tokens dynamically to different expert networks, potentially fracturing the global attention topology.

**Signal Attenuation.** We observe a distinct "Sparsity Penalty." While the spectral signal remains statistically significant ($p < 10^{-10}$), the effect size attenuates from $d \approx 3.0$ (dense) to $d \approx 1.6$ (sparse) on balanced data. We hypothesize that expert routing introduces topological noise, acting as a diffractor that scatters the spectral energy of the attention graph.

**Entropy as a Routing Proxy.** The dominant metric shifts to Spectral Entropy (Accuracy 83.0% at Layer 14). Valid proofs exhibit significantly lower entropy ($\mu = 2.17$) than invalid ones ($\mu = 2.52$). In an MoE context, this suggests that valid

reasoning corresponds to "focused routing", coherent attention to relevant contexts, whereas hallucinations manifest as "routing entropy."

**Error Agnosticism.** A taxonomic breakdown reveals that the spectral signature is remarkably error-agnostic. The effect size for detecting *Logic Errors* ($d = 0.56$) is nearly identical to that for *Calculation Errors* ($d = 0.53$), showing that even minor arithmetic deviations induce measurable geometric stress in the attention graph.

| Setting | Best Metric | Layer | $p_{\text{MW}}$ | $|d|$ | Acc. |
|---|---|---|---|---|---|
| *Qwen-MoE on MiniF2F (Balanced, $n = 100$)* | | | | | |
| | Entropy | L14 | $1.03 \times 10^{-10}$ | 1.67 | 83.0% |
| | HFER | L22 | $1.29 \times 10^{-10}$ | 1.65 | 81.0% |
| | Smoothness | L12 | $1.80 \times 10^{-9}$ | 1.50 | 83.0% |
| *Qwen-MoE on MiniF2F (Full, $n = 454$)* | | | | | |
| | Smoothness | L6 | $3.16 \times 10^{-67}$ | 2.73 | 93.6% |
| | HFER | L22 | $2.02 \times 10^{-65}$ | 2.54 | 90.5% |

*Table 22.* **Mixture-of-Experts Results.** On balanced data, effect sizes attenuate to $d \approx 1.6$ ("sparsity penalty"). On the full dataset, the signal recovers to $d = 2.73$.

### D.3. Cognitive Interpretation

The asymmetry between false positive and false negative errors suggests a cognitive interpretation of the spectral signature:

- **Invalid proofs with low spectral energy** are "confidently wrong", the model processes them smoothly despite their incorrectness, analogous to the Dunning-Kruger effect in human cognition.

- **Valid proofs with high spectral energy** involve genuine cognitive effort, the model recognizes difficulty and allocates more computational resources, producing higher-frequency activity.

Under this interpretation, the spectral signature reflects not just validity, but the model's **epistemic state**, its implicit certainty about its own reasoning. Low HFER indicates confident processing (justified for valid proofs, unjustified for "confidently wrong" invalid proofs); high HFER indicates effortful processing.

This interpretation suggests applications beyond validity classification:

- **Difficulty estimation:** High-energy valid proofs may indicate genuinely challenging problems.

- **Overconfidence detection:** Low-energy invalid proofs may signal dangerous overconfidence.

- **Uncertainty quantification:** Spectral features could complement or replace learned uncertainty estimates.

# E. Complete Results Tables

## E.1. Label Correction Statistics

| Model | Family | Orig. Valid | Corr. Valid | Reclaimed | % Change |
|---|---|---|---|---|---|
| Llama-3.2-1B | Meta | 154 | 205 | +51 | +33.1% |
| Llama-3.2-3B | Meta | 154 | 195 | +41 | +26.6% |
| Llama-3.1-8B | Meta | 154 | 193 | +39 | +25.3% |
| Qwen2.5-0.5B | Alibaba | 154 | 194 | +40 | +26.0% |
| Qwen2.5-7B | Alibaba | 154 | 187 | +33 | +21.4% |
| Phi-3.5-mini | Microsoft | 154 | 205 | +51 | +33.1% |
| Mistral-7B | Mistral AI | 154 | 190 | +36 | +23.4% |

*Table 23.* Label correction statistics across all models. All seven models independently identify 33–51 compiler-rejected proofs as spectrally valid.

# F. Reproducibility Checklist

**Code Availability.** The complete implementation including attention extraction, spectral computation, threshold optimization, and Lanczos-accelerated eigendecomposition is available at https://github.com/vcnoel/geometry-of-reason.

**Data Availability.** We use the publicly available MiniF2F benchmark (Zheng et al., 2022). Our 454-proof evaluation subset with all labels (original and corrected) and metadata will be released.

**Compute Requirements.** All experiments can be reproduced on a single NVIDIA A100 GPU (40GB) in under 3 hours total. Memory requirements are dominated by model inference, not spectral computation.

**Hyperparameters.** The only hyperparameter is the classification threshold $\tau$. We report results using both full-data calibrated thresholds and nested cross-validation. Threshold robustness analysis (Section C.5) confirms low sensitivity.

**Statistical Reporting.** All $p$-values are reported using both non-parametric Mann-Whitney U tests ($p_{\mathrm{MW}}$) and parametric two-sided Welch's $t$-tests ($p_t$) for full transparency and robustness. We apply Benjamini-Hochberg correction for multiple comparisons when scanning over layer-metric combinations. Effect sizes use Cohen's $d$ with pooled standard deviation.

**Random Seeds.** No random seeds are required: our method is fully deterministic given model weights.

## F.1. Top 10 Discriminators per Model on MiniF2F (Corrected Labels)

| Metric | Layer | $p_{\mathrm{MW}}$ | $p_t$ | Cohen's $d$ | Valid $\mu$ | Invalid $\mu$ |
|---|---|---|---|---|---|---|
| | | **Llama-3.1-8B-Instruct** (193 Valid, 261 Invalid) | | | | |
| HFER | L0 | $7.94 \times 10^{-65}$ | $2.43 \times 10^{-93}$ | $-2.88$ | 0.193 | 0.327 |
| Smoothness | L10 | $3.31 \times 10^{-64}$ | $2.03 \times 10^{-92}$ | $-2.69$ | 0.977 | 1.005 |
| Smoothness | L15 | $6.67 \times 10^{-64}$ | $5.53 \times 10^{-80}$ | $-2.11$ | 0.967 | 0.997 |
| Smoothness | L9 | $7.82 \times 10^{-64}$ | $2.26 \times 10^{-92}$ | $-2.82$ | 0.974 | 1.003 |
| HFER | L31 | $9.29 \times 10^{-64}$ | $2.45 \times 10^{-94}$ | $-2.56$ | 0.305 | 0.448 |
| HFER | L30 | $9.40 \times 10^{-64}$ | $5.44 \times 10^{-105}$ | $-3.00$ | 0.170 | 0.333 |
| Smoothness | L12 | $1.16 \times 10^{-63}$ | $8.41 \times 10^{-84}$ | $-2.63$ | 0.963 | 0.988 |
| Smoothness | L11 | $1.86 \times 10^{-63}$ | $3.91 \times 10^{-86}$ | $-2.73$ | 0.972 | 0.999 |
| Smoothness | L8 | $3.11 \times 10^{-63}$ | $1.04 \times 10^{-88}$ | $-2.74$ | 0.974 | 1.003 |
| Smoothness | L13 | $3.19 \times 10^{-63}$ | $1.85 \times 10^{-74}$ | $-2.22$ | 0.964 | 0.983 |

*Table 24.* **Top 10 discriminators: Llama-3.1-8B-Instruct.** Llama-3.1-8B-Instruct consistently favor HFER and Smoothness at mid-to-late layers, with effect sizes $|d| \geq 2.88$.

| Metric | Layer | $p_{MW}$ | $p_t$ | Cohen's $d$ | Valid $\mu$ | Invalid $\mu$ |
|---|---|---|---|---|---|---|
| | | **Llama-3.2-1B-Instruct** (205 Valid, 249 Invalid) | | | | |
| HFER | L15 | $5.73 \times 10^{-67}$ | $1.01 \times 10^{-100}$ | $-2.66$ | 0.332 | 0.450 |
| Smoothness | L7 | $8.79 \times 10^{-65}$ | $9.42 \times 10^{-79}$ | $-2.65$ | 0.963 | 0.986 |
| Entropy | L15 | $1.77 \times 10^{-64}$ | $2.34 \times 10^{-88}$ | $+2.62$ | 3.007 | 2.550 |
| HFER | L11 | $2.02 \times 10^{-64}$ | $7.04 \times 10^{-103}$ | $-2.69$ | 0.051 | 0.107 |
| Smoothness | L8 | $2.31 \times 10^{-64}$ | $3.63 \times 10^{-81}$ | $-2.52$ | 0.960 | 0.981 |
| Energy | L9 | $5.14 \times 10^{-64}$ | $1.74 \times 10^{-85}$ | $-2.66$ | 608507 | 626347 |
| Fiedler | L0 | $1.47 \times 10^{-63}$ | $1.83 \times 10^{-92}$ | $+3.02$ | 0.536 | 0.412 |
| HFER | L0 | $1.62 \times 10^{-63}$ | $2.60 \times 10^{-110}$ | $-3.00$ | 0.158 | 0.297 |
| Smoothness | L6 | $3.27 \times 10^{-63}$ | $1.07 \times 10^{-81}$ | $-2.75$ | 0.970 | 0.999 |
| Smoothness | L0 | $3.79 \times 10^{-63}$ | $7.07 \times 10^{-105}$ | $+2.70$ | 0.870 | 0.815 |
| | | **Llama-3.2-3B-Instruct** (195 Valid, 259 Invalid) | | | | |
| HFER | L0 | $3.16 \times 10^{-63}$ | $1.85 \times 10^{-101}$ | $-2.88$ | 0.233 | 0.448 |
| HFER | L11 | $3.66 \times 10^{-62}$ | $6.06 \times 10^{-102}$ | $-2.97$ | 0.106 | 0.187 |
| HFER | L26 | $6.45 \times 10^{-62}$ | $4.50 \times 10^{-106}$ | $-2.75$ | 0.126 | 0.273 |
| Smoothness | L11 | $9.83 \times 10^{-62}$ | $3.39 \times 10^{-83}$ | $-2.56$ | 0.969 | 0.999 |
| Smoothness | L9 | $2.00 \times 10^{-61}$ | $1.52 \times 10^{-89}$ | $-2.73$ | 0.978 | 1.012 |
| Smoothness | L8 | $2.07 \times 10^{-61}$ | $1.82 \times 10^{-90}$ | $-2.70$ | 0.975 | 1.006 |
| Entropy | L25 | $3.86 \times 10^{-61}$ | $1.11 \times 10^{-92}$ | $-2.84$ | 1.824 | 2.450 |
| Entropy | L27 | $3.90 \times 10^{-61}$ | $2.39 \times 10^{-92}$ | $+2.56$ | 3.026 | 2.437 |
| Smoothness | L10 | $4.56 \times 10^{-61}$ | $2.70 \times 10^{-80}$ | $-2.60$ | 0.973 | 1.001 |
| Smoothness | L0 | $6.60 \times 10^{-61}$ | $3.49 \times 10^{-89}$ | $+2.39$ | 0.794 | 0.721 |

*Table 25.* **Top 10 discriminators: Meta Llama family.** The Llama models consistently favor HFER and Smoothness at mid-to-late layers, with effect sizes $|d| \geq 2.66$. Llama-1B uniquely shows strong Fiedler discrimination at L0 ($d = 3.02$).

| Metric | Layer | $p_{\text{MW}}$ | $p_t$ | Cohen's $d$ | Valid $\mu$ | Invalid $\mu$ |
|---|---|---|---|---|---|---|
| **Qwen2.5-0.5B-Instruct** (194 Valid, 260 Invalid) | | | | | | |
| Entropy | L0 | $4.45 \times 10^{-65}$ | $1.43 \times 10^{-116}$ | +2.93 | 3.317 | 2.401 |
| Smoothness | L3 | $3.84 \times 10^{-64}$ | $2.05 \times 10^{-53}$ | −2.22 | 0.951 | 0.964 |
| Energy | L4 | $1.75 \times 10^{-63}$ | $1.28 \times 10^{-81}$ | −2.75 | 1553410 | 1598135 |
| Smoothness | L14 | $5.78 \times 10^{-63}$ | $7.43 \times 10^{-81}$ | −2.70 | 0.931 | 0.965 |
| Smoothness | L15 | $7.11 \times 10^{-63}$ | $2.02 \times 10^{-72}$ | −2.19 | 0.939 | 0.959 |
| Entropy | L1 | $1.14 \times 10^{-62}$ | $1.21 \times 10^{-96}$ | +2.88 | 2.718 | 1.839 |
| Energy | L16 | $1.87 \times 10^{-62}$ | $1.86 \times 10^{-101}$ | −2.70 | 1738815 | 1876539 |
| Energy | L19 | $1.94 \times 10^{-62}$ | $7.15 \times 10^{-101}$ | −2.82 | 1742982 | 1945174 |
| Smoothness | L4 | $2.02 \times 10^{-62}$ | $7.30 \times 10^{-59}$ | −2.10 | 0.939 | 0.954 |
| Energy | L17 | $3.65 \times 10^{-62}$ | $7.48 \times 10^{-101}$ | −2.72 | 1717977 | 1874831 |
| **Qwen2.5-7B-Instruct** (187 Valid, 267 Invalid) | | | | | | |
| HFER | L26 | $5.68 \times 10^{-55}$ | $2.45 \times 10^{-75}$ | −2.43 | 0.236 | 0.405 |
| Entropy | L2 | $9.59 \times 10^{-55}$ | $7.43 \times 10^{-79}$ | +2.38 | 2.449 | 1.845 |
| HFER | L12 | $2.17 \times 10^{-54}$ | $6.85 \times 10^{-74}$ | −1.94 | 0.073 | 0.105 |
| Entropy | L1 | $7.45 \times 10^{-54}$ | $2.05 \times 10^{-68}$ | +2.28 | 1.932 | 1.147 |
| Smoothness | L23 | $7.62 \times 10^{-54}$ | $4.66 \times 10^{-83}$ | +2.21 | 0.752 | 0.560 |
| Energy | L23 | $9.44 \times 10^{-54}$ | $1.95 \times 10^{-80}$ | +2.13 | 150401749 | 116500534 |
| Smoothness | L24 | $1.34 \times 10^{-53}$ | $2.24 \times 10^{-86}$ | +2.33 | 0.689 | 0.396 |
| Energy | L24 | $1.43 \times 10^{-53}$ | $2.59 \times 10^{-84}$ | +2.26 | 137099662 | 83429886 |
| HFER | L9 | $1.46 \times 10^{-53}$ | $8.21 \times 10^{-67}$ | −1.79 | 0.069 | 0.105 |
| Smoothness | L14 | $1.58 \times 10^{-53}$ | $4.80 \times 10^{-83}$ | +2.17 | 0.887 | 0.833 |

*Table 26.* **Top 10 discriminators: Alibaba Qwen family.** Qwen-0.5B uniquely favors Spectral Entropy at early layers ($d = 2.93$ at L0), achieving $p_t = 1.43 \times 10^{-116}$, the most significant result in our study. Qwen-7B shows a more diverse metric profile with HFER, Entropy, and Smoothness all contributing.

| Metric | Layer | $p_{\text{MW}}$ | $p_t$ | Cohen's $d$ | Valid $\mu$ | Invalid $\mu$ |
|---|---|---|---|---|---|---|
| **Phi-3.5-mini-Instruct** (205 Valid, 249 Invalid) | | | | | | |
| Smoothness | L27 | $1.69 \times 10^{-66}$ | $5.60 \times 10^{-100}$ | $+3.21$ | $0.496$ | $0.239$ |
| Smoothness | L26 | $2.37 \times 10^{-66}$ | $1.86 \times 10^{-103}$ | $+3.27$ | $0.464$ | $0.153$ |
| Smoothness | L25 | $4.51 \times 10^{-66}$ | $2.33 \times 10^{-107}$ | $+3.30$ | $0.438$ | $0.076$ |
| Smoothness | L28 | $4.74 \times 10^{-66}$ | $2.03 \times 10^{-95}$ | $+3.12$ | $0.543$ | $0.343$ |
| HFER | L26 | $5.64 \times 10^{-66}$ | $2.41 \times 10^{-95}$ | $-3.16$ | $0.312$ | $0.560$ |
| Energy | L26 | $9.95 \times 10^{-66}$ | $5.86 \times 10^{-75}$ | $+2.28$ | $21029690$ | $10964297$ |
| Smoothness | L23 | $2.64 \times 10^{-65}$ | $4.34 \times 10^{-113}$ | $+3.29$ | $0.395$ | $-0.095$ |
| Smoothness | L24 | $3.29 \times 10^{-65}$ | $6.49 \times 10^{-111}$ | $+3.30$ | $0.418$ | $-0.012$ |
| Energy | L25 | $4.10 \times 10^{-65}$ | $6.39 \times 10^{-101}$ | $+2.79$ | $18433282$ | $4584151$ |
| Smoothness | L22 | $6.16 \times 10^{-65}$ | $3.54 \times 10^{-114}$ | $+3.28$ | $0.384$ | $-0.144$ |
| **Mistral-7B-v0.1-Instruct** (190 Valid, 264 Invalid) | | | | | | |
| HFER | L11 | $3.94 \times 10^{-49}$ | $1.10 \times 10^{-52}$ | $-1.57$ | $0.121$ | $0.145$ |
| Smoothness | L26 | $1.16 \times 10^{-48}$ | $1.21 \times 10^{-78}$ | $+2.09$ | $0.639$ | $0.354$ |
| HFER | L30 | $2.85 \times 10^{-48}$ | $3.78 \times 10^{-67}$ | $-1.87$ | $0.234$ | $0.313$ |
| Smoothness | L25 | $3.45 \times 10^{-48}$ | $1.25 \times 10^{-77}$ | $+2.07$ | $0.627$ | $0.327$ |
| Smoothness | L24 | $6.31 \times 10^{-48}$ | $4.07 \times 10^{-77}$ | $+2.05$ | $0.630$ | $0.334$ |
| Energy | L25 | $2.60 \times 10^{-47}$ | $1.72 \times 10^{-69}$ | $+1.86$ | $128639$ | $81926$ |
| Smoothness | L27 | $3.38 \times 10^{-47}$ | $3.02 \times 10^{-75}$ | $+2.04$ | $0.640$ | $0.360$ |
| Energy | L24 | $4.63 \times 10^{-47}$ | $1.77 \times 10^{-69}$ | $+1.86$ | $125535$ | $79445$ |
| Smoothness | L23 | $6.02 \times 10^{-47}$ | $1.49 \times 10^{-74}$ | $+2.00$ | $0.674$ | $0.415$ |
| Energy | L26 | $6.08 \times 10^{-47}$ | $1.37 \times 10^{-64}$ | $+1.76$ | $135930$ | $95322$ |

*Table 27.* **Top 10 discriminators: Microsoft Phi and Mistral AI.** Phi-3.5-mini exhibits the largest effect sizes in our study ($d = 3.30$) via late-layer Smoothness, with invalid proofs showing negative smoothness (L22–L24). Mistral-7B's Sliding Window Attention shifts the optimal metric from HFER ($d = 1.57$) to Smoothness ($d = 2.09$), demonstrating that attention mechanism architecture determines which spectral features best capture reasoning validity.

| Metric | Layer | $p_{\mathrm{MW}}$ | $p_t$ | Cohen's $d$ | Valid $\mu$ | Invalid $\mu$ |
|---|---|---|---|---|---|---|
| **Qwen-MoE-A2.7B** (194 Valid, 260 Invalid) | | | | | | |
| Smoothness | L13 | $2.79 \times 10^{-67}$ | $2.49 \times 10^{-84}$ | $-2.19$ | 0.931 | 0.971 |
| Smoothness | L6 | $3.16 \times 10^{-67}$ | $4.34 \times 10^{-92}$ | $-2.73$ | 0.937 | 0.980 |
| Smoothness | L12 | $6.22 \times 10^{-67}$ | $5.47 \times 10^{-83}$ | $-2.15$ | 0.942 | 0.995 |
| Smoothness | L11 | $1.92 \times 10^{-66}$ | $4.22 \times 10^{-88}$ | $-2.25$ | 0.950 | 1.003 |
| Smoothness | L8 | $2.20 \times 10^{-66}$ | $1.03 \times 10^{-87}$ | $-2.25$ | 0.940 | 0.986 |
| Smoothness | L7 | $3.24 \times 10^{-66}$ | $1.45 \times 10^{-86}$ | $-2.38$ | 0.941 | 0.972 |
| Smoothness | L14 | $1.14 \times 10^{-65}$ | $1.93 \times 10^{-94}$ | $-2.39$ | 0.949 | 1.006 |
| HFER | L22 | $2.02 \times 10^{-65}$ | $1.68 \times 10^{-98}$ | $-2.54$ | 0.272 | 0.378 |
| Smoothness | L10 | $2.82 \times 10^{-65}$ | $2.21 \times 10^{-81}$ | $-2.13$ | 0.943 | 0.989 |
| Smoothness | L9 | $4.45 \times 10^{-65}$ | $3.38 \times 10^{-86}$ | $-2.23$ | 0.943 | 0.990 |

*Table 28.* **Top 10 discriminators: Qwen-MoE-A2.7B.** Smoothness dominates across early-to-mid layers ($d$ up to 2.73 at L6), with HFER providing complementary discrimination at L22 ($d = 2.54$). Effect sizes are attenuated relative to dense models, consistent with the "sparsity penalty" reported in Section D.2.

## G. Ethical Considerations

The authors identify no significant ethical risks associated with the mechanistic probing techniques presented in this work. Regarding the preparation of this manuscript, Large Language Models (specifically Gemini 3.0 Pro) were used to assist with prose refinement, grammatical cleaning, and code refactoring. All outputs were manually reviewed and verified by the authors, who take full responsibility for the final content and technical accuracy of the paper.

## H. Every layer plots

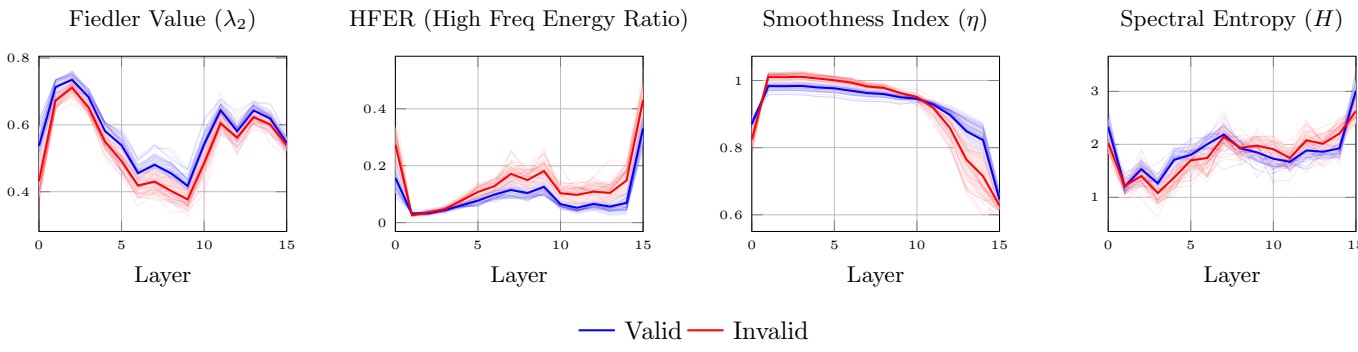

*Figure 6.* Layer-wise spectral metrics (Combined) for Llama 3.2 1B Instruct.

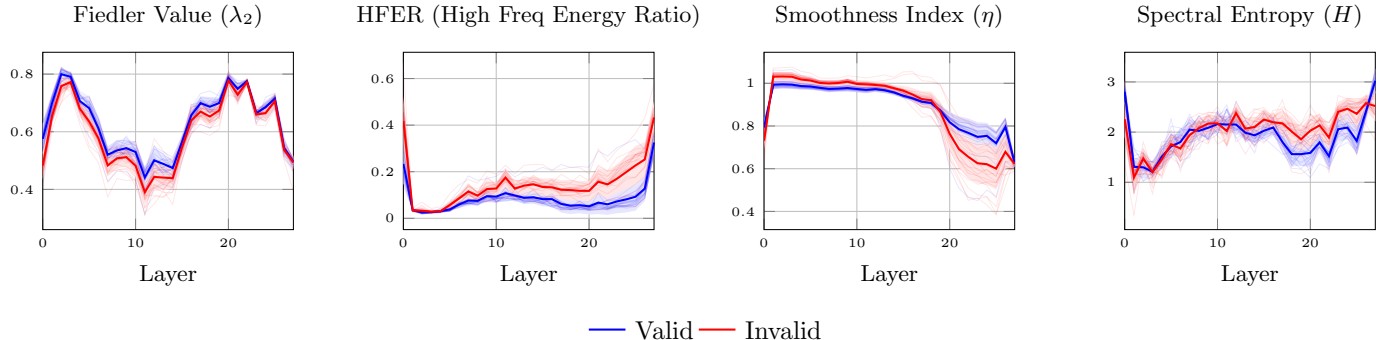

*Figure 7.* Layer-wise spectral metrics (Combined) for Llama 3.2 3B Instruct.

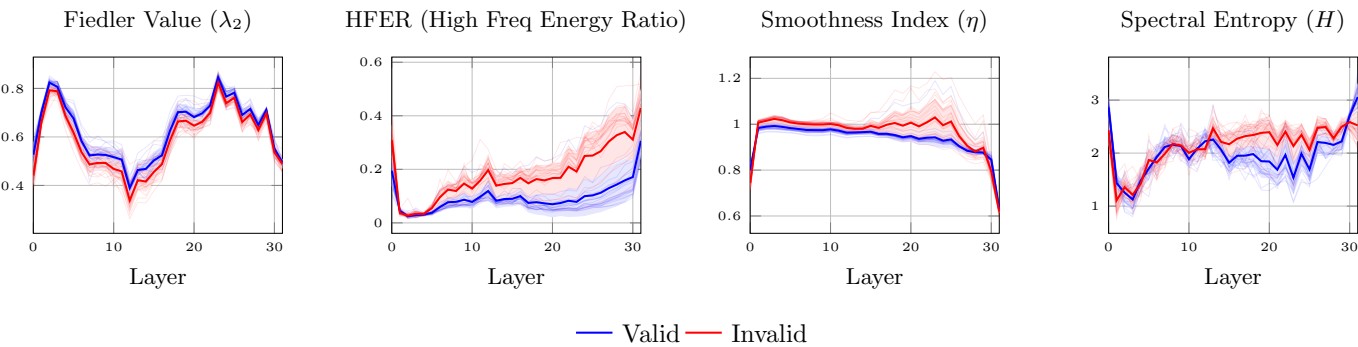

*Figure 8.* Layer-wise spectral metrics (Combined) for Llama 3.1 8B Instruct.

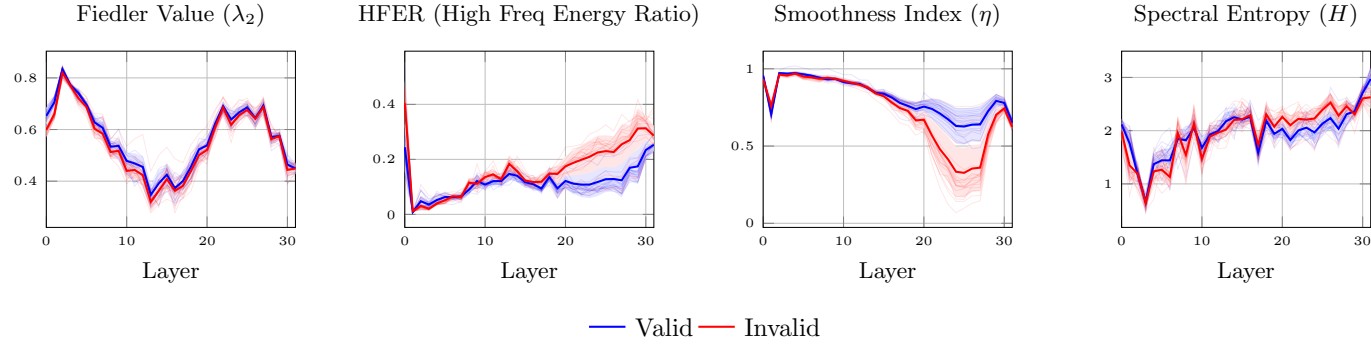

*Figure 9.* Layer-wise spectral metrics (Combined) for Mistral 7B.

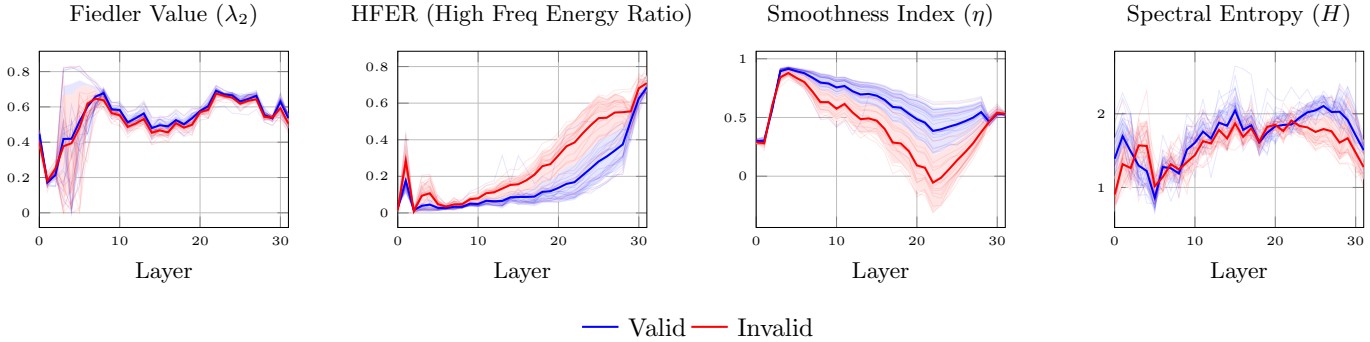

*Figure 10.* Layer-wise spectral metrics (Combined) for Phi 3.5 Mini Instruct.

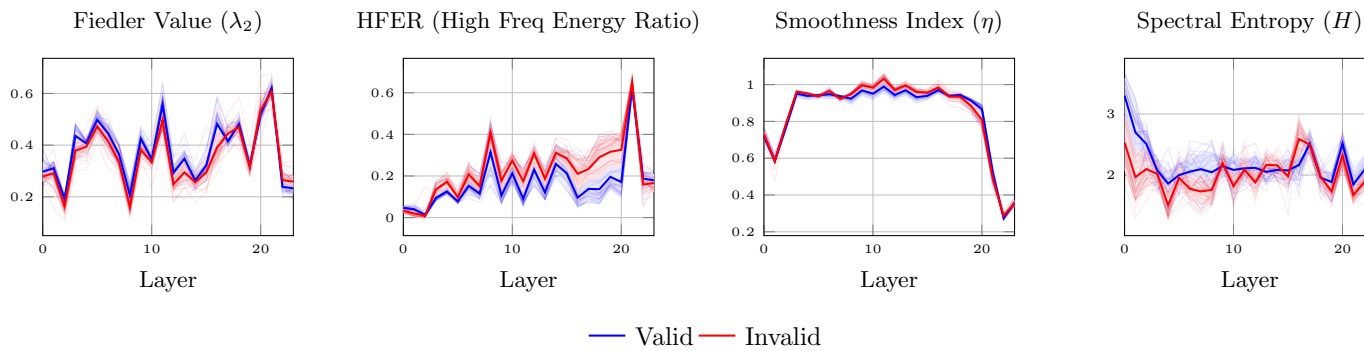

*Figure 11.* Layer-wise spectral metrics (Combined) for Qwen 2.5 0.5B Instruct.

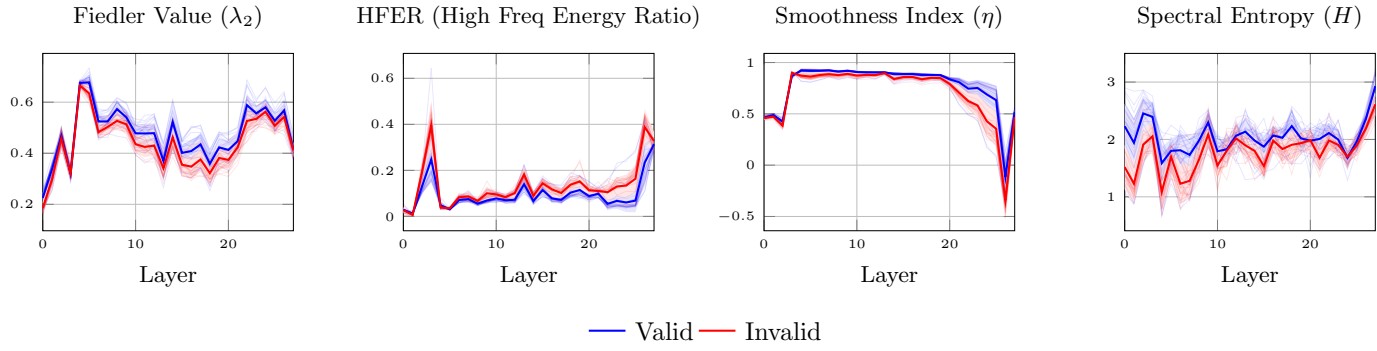

*Figure 12.* Layer-wise spectral metrics (Combined) for Qwen 2.5 7B Instruct.

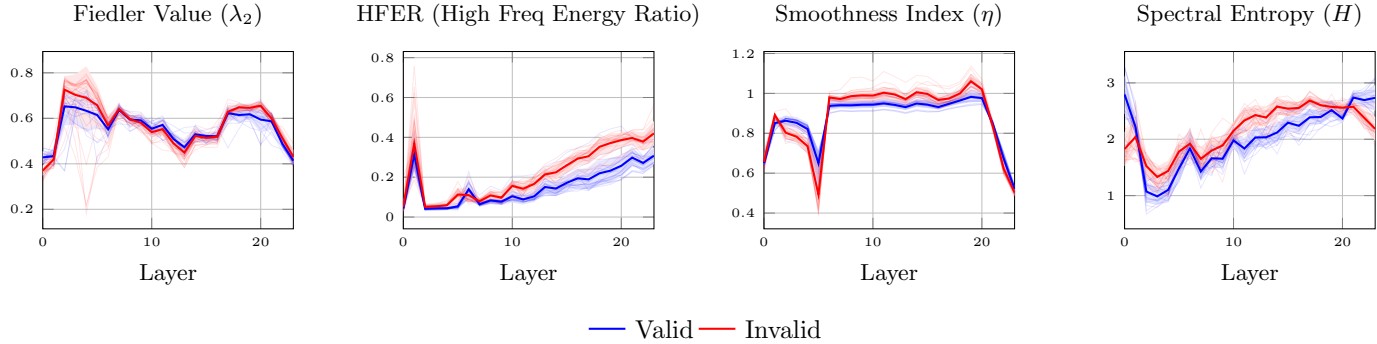

*Figure 13.* Layer-wise spectral metrics (Combined) for Qwen 1.5 MoE A2.7b Chat.

