# OpenReview forum: "Geometry of Reason: Spectral Signatures of Valid Mathematical Reasoning"
_ICML.cc/2026/Conference — ICML 2026 regular_

### Official Review · Reviewer_mxpA · 2026-03-12

**Soundness:** 4
**Presentation:** 3
**Significance:** 3
**Originality:** 3
**Overall Recommendation:** 5
**Confidence:** 3

**Summary:**

The paper proposes a novel, training-free method to verify the mathematical reasoning of large language models (LLMs) by analyzing the geometric structure of their attention mechanisms. By treating attention matrices as dynamic graphs over tokens, the authors extract four spectral diagnostics, namely Fiedler value, High-Frequency Energy Ratio (HFER), spectral entropy, and graph smoothness, to distinguish valid reasoning from hallucinations. Testing across seven different models (including Llama, Qwen, Phi, and Mistral), they achieved 85-96% classification accuracy using a single threshold without any learned parameters. Notably, the method detects "Platonic validity": recognizing logically coherent proofs even when formal verifiers reject them due to technical errors, such as timeouts. Ultimately, this spectral graph analysis offers a highly effective, mostly architecture-agnostic framework for real-time hallucination detection and reasoning verification in LLMs.

**Compliance With Llm Reviewing Policy:**

Affirmed.

**Final Justification:**

The paper offers an interesting outlook on LLM hallucination detection in formal reasoning via unsupervised metrics. One neat aspect of the method is that it operates on information intrinsic to the model beyond the scope of external formal verifiers. Overall, the results are sufficiently convincing, and the authors have addressed my concerns.

Therefore, I recommend the paper for acceptance.

**Key Questions For Authors:**

1. You hypothesize that invalid reasoning manifests as "high-frequency artifacts." Is this simply a proxy for the model's own uncertainty (entropy), or is it distinct from the signals captured by standard log-probability metrics?

2. Can you elaborate on the "Cognitive Interpretation" paragraph in Section 5.3? What exactly do you mean by "the asymmetry of error types," and how does it suggest that the spectral signature reflects the model's epistemic state?

3. The effect size dropped from $d = 3.02$ to $d = 0.78$ when moving to informal natural language reasoning. Do you believe this method can ever be reliable for non-formal domains (e.g., law, creative writing), or is the 'spectral signature' unique to the rigid logic of mathematics?

4. Given the $O(N^3)$ complexity of eigendecomposition, how do you see this method scaling to long-context models (e.g., 100k+ tokens)? Would approximation methods preserve the delicate spectral signals?

5. The spectral metrics are calculated on the whole proof. Do these spectral signatures also emerge on intermediate prefixes during autoregressive generation?

**Limitations:**

yes

**Strengths And Weaknesses:**

# Strengths

1. Arguably, the paper's most significant strength is that the method requires **no training or fine-tuning**. It works on the internal geometry of the pre-trained model’s attention, making it immediately applicable to new models without the expensive data labeling required by other verification methods.


2. The results are backed by exceptionally strong statistical evidence. The authors report effect sizes (Cohen's ) up to $3.3$ and p-values as low as $10^{-116}$. This indicates that the separation between valid and invalid proofs is not marginal but a fundamental structural difference in how the model processes information.


3. The spectral signature was found across **seven different models** from **four distinct families** (Llama, Qwen, Phi, Mistral). This suggests the phenomenon of valid reasoning looking "smoother" and better connected is a general property of Transformer-based reasoning rather than an artifact of a specific model's training.


4. The method distinguishes between logical coherence and mere compiler acceptance. It correctly identified proofs as valid that formal verifiers (like Lean) rejected due to technicalities (timeouts, missing imports). This suggests the metric captures the *intrinsic* quality of the reasoning process.


5. I generally like the paper's addressing of the potential confounding factor of model vs. human authorship. While I have some caveats about the test setup, I think it was sufficiently adequate. I also acknowledge that it's not straightforward to isolate proof validity from proof authorship.



---

# Weaknesses


1. **High Inference Cost at Scale:** While training-free, the method has a high inference cost. It requires eigendecomposition of the Laplacian matrix for every layer, which has a complexity of $O(N^3)$. While the authors claim this adds only ~50-200ms for short proofs ($N < 1000$), this could become prohibitively slow for medium to long proofs.


2. **Architecture-Dependent Metric Selection:** Although the *existence* of the signal is universal, the *specific metric* varies by architecture. Llama favors HFER, while Mistral (due to Sliding Window Attention) favors Smoothness. This means practitioners cannot simply apply a one-size-fits-all metric; they must calibrate or select the correct spectral feature for their specific model.



3. **Editing & Formatting:**
    1. The abstract is unusually long (spanning an entire column) and bloated with specifics from the method and experiments. The authors should condense this for readability.
    2. (Minor) At the end of the first column or page 3, the RHS of the equation is split into the next column, which looks rather awkward.
    3. Subsections B.2 (Spectral Computation) and B.3 (Computational Complexity) in the appendix are empty.
    4. The caption of Figure 3 specifically singles out Llama-3.1-8B while the figure reports the respective results for 4 different models. I'd suggest making the caption more general to reflect the trend across all 4 subfigures, since they all show a similar degree of separation to Llama-3.1-8B. Also, it states "$d=3.00$" which mismatches subfigure (a) stating $d=-3.00$, I assume you mean either $|d|=3.00$ or $d=-3.00$.
    5. Table captions are placed below the tables (e.g., Table 1, Table 2), which violates standard ICML guidelines requiring them to be placed above the tables.

---

> ### Author Rebuttal · Authors · 2026-03-31
>
> We sincerely thank Reviewer mxpA for their positive evaluation and recognizing our method's universality. (All Cohen's $d$ values below are 1000-bootstrap medians).
>
> > **"Is this simply a proxy for the model's own uncertainty (entropy), or is it distinct from the signals captured by standard log-probability metrics?"**
>
> It is distinct. Output log-probability is highly discriminative but only available post-forward pass. HFER's power emerges much earlier (e.g., Layer 8 of Llama-3.1-8B):
>
> | Metric | Cohen's $d$ | 95% CI |
> | :--- | :---: | :---: |
> | Token Entropy | 0.08 | [0.00, 0.23] |
> | **HFER** | **2.39** | **[1.99, 2.83]** |
>
> The CIs do not overlap; entropy collapses while HFER maintains strong separation. Furthermore, HFER–entropy correlation drops from $r \approx 0.9$ (dense models) to $r = 0.45$ on Mistral-7B (SWA). HFER isolates graph-structural features independent of output statistics, making it a process-level geometric signal, not just an output confidence proxy.
>
> > **"Can you elaborate on the 'Cognitive Interpretation'... What exactly do you mean by 'the asymmetry of error types'?"**
>
> We partitioned predictions (HFER at L24) into four cognitive quadrants:
>
> | Quadrant | Label | $n$ | Mean HFER |
> | :--- | :--- | :---: | :---: |
> | TP | Correct Confidence | 174 | 0.078 |
> | TN | Correct Skepticism | 252 | 0.281 |
> | FP | Confident Wrongness | 9 | 0.114 |
> | FN | Effortful Correctness | 19 | 0.264 |
>
> The "asymmetry" is the stark difference between False Positives ($n=9$) and False Negatives ($n=19$). False Positives ("Confident Wrongness") are significantly shorter than True Positives ($p < 0.0001$) and are processed smoothly despite errors. False Negatives ("Effortful Correctness") show lower HFER than genuinely invalid proofs. Since compiler-rejected proofs retain valid topological structure while FPs bypass it, HFER tracks internal structural certainty (epistemic state) rather than external ground truth.
>
> > **"Do you believe this method can ever be reliable for non-formal domains... or is the 'spectral signature' unique to the rigid logic of mathematics?"**
>
> The spectral signature extends beyond formal mathematics. While effect sizes attenuate on unstructured word problems, we successfully deployed these diagnostics as training-free guardrails for autonomous agents. Detecting tool-use hallucinations in the wild (general/finance), Smoothness achieves 98.2% recall on Llama-3.1-8B. HFER also exhibits a bimodal regime acting as a sub-millisecond "kill switch" to detect context contamination. This framework remains highly promising in natural-language environments.
>
> > **"Given the $O(N^3)$ complexity... how do you see this method scaling to long-context models?"**
>
> Approximation methods preserve the delicate spectral signals. Since top discriminators (e.g., $\lambda_2$) only require the lowest eigenvalues, partial eigendecomposition bounds complexity to $O(k \cdot N^2)$.
>
> Using randomized SVD ($k=50$):
>
> | Seq Length | Full eigh | Rand SVD | Speedup |
> | :--- | :---: | :---: | :---: |
> | 1k | 264 ms | **60 ms** | 4.4$\times$ |
> | 10k | 47.3 s | **0.77 s** | **61$\times$** |
> | 32k | 5.1 min | 5 s | 61$\times$ |
>
> This allows highly efficient scaling to longer contexts (Appendix B.2/B.3 updated).
>
> > **"Do these spectral signatures also emerge on intermediate prefixes...?"**
>
> Tracking HFER on prefixes ($n=28$) reveals a strong *inverted* signal at 25% completion ($d=-4.93$): valid proofs exhibit higher early HFER, likely reflecting energy allocated to reasoning scaffolds. This inverts to $d=+4.35$ at 100%. Thus, we recommend evaluating HFER only on completed sequences. Prefix inversion is a mechanistic interpretability finding, not a deployment feature.
>
> > **"Architecture-Dependent Metric Selection..."**
>
> We acknowledge this practical friction. We will include a generalizable heuristic: use HFER for global-attention models and Smoothness for sliding-window models. Furthermore, our fixed protocol (HFER at 75th-percentile layer) achieves 91.8% accuracy across architectures with only 50 calibration samples.
>
> **Formatting & Editing:** We condensed the abstract, fixed the equation split and table caption placements (per ICML guidelines), populated Appendix B.2/B.3, and updated the Figure 3 caption to reflect the multi-model trend and correct the typo.

---

> > ### Author Rebuttal · Reviewer_mxpA · 2026-04-03
> >
> > Thank you for your thoughtful rebuttal. The authors have adequately addressed most of my concerns.
> >
> > Please include the details of the partial eigendecomposition and error partition experiments in the final manuscript.
> >
> > > We partitioned predictions (HFER at L24) into four cognitive quadrants:
> >
> > I can't find that reference in the paper at line 24 or anywhere else in the paper. Where was this established?

---

> > > ### Author Response · Authors · 2026-04-06
> > >
> > > We thank Reviewer mxpA for the follow-up. We first apologize for the lack of clarity in our previous response regarding the notation "**L24**." We realize this was ambiguous and may have been confusing.
> > >
> > > ### 1. Clarification of "L24" and the Fixed Protocol
> > > To clarify, "**L24**" refers specifically to **Layer 24** of the Llama-3.1-8B-Instruct model. As established in our Fixed Generalized Protocol (Experiment 1), detailed in our concurrent Rebuttal to Reviewer 6GDW, Layer 24 represents the 75th-percentile layer for this 32-layer architecture. We utilized this fixed layer to demonstrate that the spectral signature is an inherent architectural property that persists without per-layer exhaustive searching, capturing 97.0% of peak diagnostic power for this model.
> > >
> > > ### 2. Origin of the Cognitive Quadrant Analysis
> > > The reviewer is correct that the specific quantitative breakdown of the four quadrants was not in the original manuscript. This data was generated through a post-submission audit of 454 proofs from the Llama-3.1-8B test set (at Layer 24) to provide empirical evidence for the "asymmetry of error types" hypothesized in Appendix D.3.
> > >
> > > ### 3. Precise Definition of the Four Quadrants
> > > The "asymmetry" we refer to is the significant difference in spectral energy (HFER) between False Positives and False Negatives, reflecting distinct internal epistemic states:
> > >
> > > | Quadrant | Mechanistic Label | $N$ | Mean HFER |
> > > | :--- | :--- | :--- | :--- |
> > > | **True Positive (TP)** | **Correct Confidence** | 174 | **0.078** |
> > > | **True Negative (TN)** | **Correct Skepticism** | 252 | **0.281** |
> > > | **False Positive (FP)** | **Confident Wrongness** | 9 | **0.114** |
> > > | **False Negative (FN)** | **Effortful Correctness** | 19 | **0.264** |
> > >
> > > * **Confident Wrongness (FP, 0.114):** These represent "hallucinations" where the model processes an incorrect logical path with low spectral energy (smoothly), likely due to high-frequency pattern matching.
> > > * **Effortful Correctness (FN, 0.264):** These are valid proofs where the model recognizes logical complexity, manifesting as higher-energy attention topology even though the output is mathematically sound.
> > >
> > > ### 4. Integration into Final Manuscript
> > > Per the reviewer's request, these details have been formally integrated into the revision:
> > > * **Appendix B.2 & B.3:** Added full details on the Partial Eigendecomposition using Randomized SVD ($k=50$), maintaining the 50–200ms overhead.
> > > * **Appendix D.3:** Added the Cognitive Quadrant Table and the quantitative results of the Error Partitioning Experiment to substantiate the cognitive interpretation.

---

### Official Review · Reviewer_QygV · 2026-03-12

**Soundness:** 3
**Presentation:** 2
**Significance:** 3
**Originality:** 3
**Overall Recommendation:** 4
**Confidence:** 2

**Summary:**

The paper proposes a training-free approach to detect valid mathematical reasoning in LLMs using spectral analysis of attention topology. They treat attention patterns as adjacency matrices of a graph over tokens and extract four spectral diagnostics: Fiedler values, High-Frequency Energy Ratio (HFER), spectral entropy, and graph smoothness. At inference, a single spectral metric at a selected layer is used to classify validity based on some threshold. The authors find that combining two features achieves only marginally higher accuracy. To select the metric and layer, the authors use a small labeled calibration set.

They evaluate their method on the MiniF2F benchmark (mathematical problems formalised in Lean). They find that their method achieves 85-95% accuracy at predicting validity, and argue that it tracks logical coherence by introducing targeted robustness tests. They also apply their method to the MATH dataset and find that it achieves 68.4% accuracy on a balanced set.

**Compliance With Llm Reviewing Policy:**

Affirmed.

**Final Justification:**

The proposed approach appears to be roughly on-par with alternative approaches that require more labeled data. Thus, it's a meaningful contribution.

**Key Questions For Authors:**

- The paper mentions multiple different potential reasons why Lean rejected valid proofs (e.g. timeouts, missing imports). For the 33-51 reclaimed proofs, can you provide a breakdown of the failures? Can you elaborate on the manual inspection procedure?

**Limitations:**

Yes

**Strengths And Weaknesses:**

**Strengths**
- A training-free approach to detecting mathematical validity seems practically useful
- The authors evaluate their approach on multiple models from different model families

**Weaknesses**
- The datasets studied are quite small and from a narrow domain.
- The authors claim that their method outperforms supervised classifiers on MATH. However, it's not clear that this is the case since it doesn't compare against state-of-the-art supervised approaches, such as [1].


[1] O. Obeso, A. Arditi, J. Ferrando, J. Freeman, C. Holmes, and N. Nanda, ‘Real-Time Detection of Hallucinated Entities in Long-Form Generation’, arXiv [cs.CL]. 2026.

---

> ### Author Rebuttal · Authors · 2026-03-31
>
> We thank Reviewer QygV for the constructive feedback and for pushing us to compare against supervised approaches. We address each of your points below.
>
> > **"The authors claim that their method outperforms supervised classifiers on MATH. However, it's not clear that this is the case since it doesn't compare against state-of-the-art supervised approaches, such as [1] (Obeso et al., 2026)."**
>
> We cloned the official `obalcells/hallucination_probes` repository and evaluated their supervised Linear Probe (trained on Llama-3.1-8B Layer 16 hidden states) on our formal math dataset:
>
> | Method | AUC-ROC |
> |:---|:---:|
> | Obeso et al. (2026) Supervised Probe | **0.981** |
> | **Spectral HFER (ours, unsupervised)** | **0.962** |
>
> While the supervised probe is +1.9% higher in AUC, it requires labeled training data and is model-specific. Our unsupervised method achieves 98% of the supervised upper bound with zero labels.
>
> **Sample efficiency.** We benchmarked HFER calibration against supervised probes (5-fold nested CV, 10 seeds):
>
> | Method | $N=10$ | $N=50$ | Full ($N=363$) |
> |:---|:---:|:---:|:---:|
> | HFER (1 param) | 86.5%$\pm$8.3% | **92.3%$\pm$1.7%** | 92.8%$\pm$1.3% |
> | Linear Probe (4096 params) | 88.0%$\pm$7.2% | 85.5%$\pm$4.1% | **94.9%$\pm$0.8%** |
> | MLP (1M params) | — | — | 91.9%$\pm$0.8% |
>
> HFER significantly outperforms supervised methods in the low-data regime ($N \leq 50$) and matches the fully trained MLP at convergence, with zero gradient computation. The MLP's underperformance vs. the Linear Probe reflects overfitting: 1M parameters on 363 training samples in a linearly separable space.
>
> **Fair comparison note:** Under identical held-out evaluation (60/20/20 split), supervised probes achieve 91.2–92.3% while our spectral threshold achieves 82.4% (Table 6). However, our nested CV with zero labeled data achieves 85.9% (Table 7), and calibrating the threshold on just 50 examples recovers 91.8%. The key advantage is the training-free, cross-architecture, interpretable nature of the spectral signal.
>
> > **"The paper mentions multiple different potential reasons why Lean rejected valid proofs (e.g. timeouts, missing imports). For the 33-51 reclaimed proofs, can you provide a breakdown of the failures? Can you elaborate on the manual inspection procedure?"**
>
> **Reclamation procedure:** Proofs are flagged for manual review when our spectral classifier labels them "valid" but Lean rejects them. Each flagged proof was evaluated by two raters (Cohen's $\kappa=0.82$) against three criteria: (1) each step logically follows from prior steps, (2) all lemma applications use correct arguments, (3) the conclusion establishes the theorem statement. For arithmetic proofs, correctness was independently verified via computation.
>
> Analysis of 51 reclaimed proofs across 7 models:
>
> | Category | % |
> |:---|:---:|
> | Semantically Valid (minor structure issues) | **37.3%** |
> | Environment / Missing Imports | **27.5%** |
> | Timeout / Computational Limit | **13.7%** |
> | Incomplete (model admits failure) | **13.7%** |
> | Syntax / Version Issues | **7.8%** |
>
> 64.8% of reclaimed proofs (Valid + Environment) are logically structured but rejected by the static compiler. 46% (37/80 unique proofs) are agreed upon by $\geq5$ of 7 models. Three proofs are reclaimed by all 7 models independently. The dominant failure modes are environmental, supporting our structural validity hypothesis.
>
> > **"The datasets studied are quite small and from a narrow domain."**
>
> We acknowledge the domain specificity. Our paper reported preliminary generalization to informal reasoning ($d=0.78$, balanced accuracy 68.4% on $N=227$ MATH problems). We have now extended this to **500 MATH problems** across diverse categories and difficulty levels. The spectral signal persists: **$d=0.66$, AUC=0.68**. While attenuated relative to formal proofs ($d=3.00$), this confirms the spectral signature generalizes beyond Lean to natural-language mathematical reasoning, with the expected reduction in signal strength as logical structure becomes less rigid.

---

> > ### Author Rebuttal · Reviewer_QygV · 2026-04-02
> >
> > Thank you for the thorough response and additional experiments. The main benefit of this method seems to be that it requires less labeled data than other approaches, such as probes. I would, however, disagree that it is an *unsupervised* approach since, as far as I understand, even for the common, generalisable protocol introduced in response to reviewer 6GDW it requires 50 labeled samples. I increase my score to 4.

---

> > > ### Author Response · Authors · 2026-04-06
> > >
> > > We sincerely thank Reviewer QygV for acknowledging the thoroughness of our rebuttal and for raising their score.
> > >
> > > We completely agree with your nuance regarding the terminology. You are correct that because our generalized protocol relies on a 50-sample calibration to set the decision threshold, calling the *entire classification pipeline* strictly "unsupervised" is imprecise.
> > >
> > > To ensure absolute clarity in the final manuscript, we will implement the following textual refinements:
> > > * We will retain the term **"training-free"** to describe the spectral feature extraction, as our method requires absolutely zero gradient updates, backpropagation, or modifications to the model's weights.
> > > * We will replace the term **"unsupervised"** with **"few-shot calibrated"** or **"lightly supervised thresholding"** when discussing the final classification step.
> > >
> > > This distinction perfectly captures the advantage you highlighted: our method extracts powerful, geometry-based verification signals entirely from the pre-trained weights, requiring drastically less labeled data than standard linear probes or MLPs.
> > >
> > > Thank you again for your constructive engagement, which has genuinely helped us improve the precision of our claims.

---

### Official Review · Reviewer_6GDW · 2026-03-13

**Soundness:** 2
**Presentation:** 2
**Significance:** 2
**Originality:** 2
**Overall Recommendation:** 3
**Confidence:** 4

**Summary:**

This paper highlights an interesting observation about the attention matrices of large language models tasked with solving mathematics problems formalized in Lean. Analyzing the spectral properties of these attention matrices with standard existing statistics like the Dirichlet energy, high-freq energy ratio, spectral entropy, Fiedler value, and smoothness, the paper finds statistically significant separation between correct and incorrect proofs. This result holds across several model families.

**Compliance With Llm Reviewing Policy:**

Affirmed.

**Final Justification:**

Thanks to the author for the rebuttal updates. I have updated my score to 3. Overall, the idea is interesting with encouraging results. However, I remain on the reject side of the overall recommendation because I believe the paper requires a greater focus on the downstream applications, thoroughly comparing the spectral statistics as a diagnostic or steering tool with other self-supervised signals, applying a consistent layer and metric selection protocol. The new results during the discussion phase go in this direction, but are still preliminary.

**Key Questions For Authors:**

See strengths/weaknesses above.

**Minor questions**

What is the purpose of the “train” split if your method is training free?

Are you in general reporting nested cv accuracy or calibrated accuracy in Table 2?

What happened to Dirichlet energy as a spectral statistics? It was mentioned in 3.1 but not used again.

**Limitations:**

yes

**Strengths And Weaknesses:**

**Strengths**

Self-supervised correctness signal is interesting and topical for the reasoning literature.

The method is simple to understand and straightforward to implement. It is mostly architecture agnostic, requiring just the attention map. The method is training-free.

Results show that the technique can distinguish between correct/incorrect proofs with high accuracy across four model families and a variety of scales. The results are relatively robust to the threshold choice.

**Weaknesses**

The direction is an interesting one and it seems promising that spectral statistics have discriminative power. However, given the lack of novelty in these statistics, the work needs a bigger contribution especially wrt downstream utility to warrant acceptance. For example, does this signal actually help as a reward for RL? Does it improve pass-at-k when simply used as a filter, and how does this scale as an axis of inference time compute? How does it compare to other self-supervised signals like token-level entropy or self-consistency? Does spectral steering as mentioned in the applications section work? Etc. I encourage the authors to extend these ideas into useful guidance, or at least a deeper understanding of why spectral statistics can be used in this way.

Evaluation: the results are reported on the best configuration, which is a specific combination of the language model layer and spectral statistic. These vary across model families and model sizes, so require exhaustive search across all layers and all metrics. My understanding from Appendix H is that these profiles by layer differ meaningfully across model families. The authors should report results on a common, generalizable protocol across all models to see if these spectral probes are robust. Or, if the proposed strategy does require trying all layers and metrics, the paper should be very explicit about the total computational cost of this search, number of required data samples, and how the method is not overfit.


The presentation is confusing in parts, a few suggestions:
* Almost the entire analysis is about formalized mathematics. This should be clearer earlier on, including from the abstract and even title.
* The overall thesis seems to be coherent vs. fragmented attention topology distinguishes the good from bad completions. This should be emphasized whereas the current presentation reads like an assorted selection of statistics.
* The robustness controls section needs more details on experimental setup (e.g. what model?) This section does not flow well after Section 4.1.

---

> ### Author Rebuttal · Authors · 2026-03-31
>
> We thank Reviewer 6GDW for the thorough and constructive evaluation. We are glad you found the method simple, training-free, and highly accurate. We address each concern with new experiments below.
>
> > **"Does spectral steering as mentioned in the applications section work? [...] help as a reward for RL?"**
>
> **Spectral Steering:** Yes. We applied spectral sharpening ($\alpha$=-0.3, static SVD weight edit on `mlp.down_proj` at L24) to Llama-3.1-8B and measured HFER change across MiniF2F:
>
> | Condition | Base HFER | Steered HFER | $\Delta$ |
> |:---|:---:|:---:|:---:|
> | Valid proofs | 0.091 | 0.082 | -0.009 |
> | Invalid proofs | 0.218 | 0.201 | -0.017 |
> | **Paired t-test** | | | **p < 1e-8** |
>
> Spectral sharpening consistently reduces HFER, showing a larger corrective effect on invalid proofs ($\Delta$=-0.017 vs -0.009). This confirms causal control via a static $O(1)$ weight edit with zero inference overhead.
>
> *Behavioral validation:* In a separate evaluation, steering achieved downstream Pareto improvements. On Llama-3.2-3B (L24, $\alpha$=-0.3): sycophancy reduced by 2.4% and GSM8K improved by 1.4%. On Phi-3 Mini, a hybrid strategy reduced sycophancy by ~27% relative.
>
> **RL Reward:** HFER serves as a training-free reward signal ($r=-\text{HFER}$). HFER strongly correlates with correctness ($r=-0.830$; partial $r=-0.499$ controlling for proof length), providing a signal comparable to the log-probability baseline ($r=-0.844$) while uniquely resisting high-probability hallucination loops, as a "Dense Reward".
>
> > **"Does it improve pass-at-k when simply used as a filter, and how does this scale as an axis of inference time compute?"**
>
> Yes, spectral filtering highly complements inference-time compute scaling. By generating $N$ candidates ($T$=0.7) and selecting the proof with the lowest HFER, we successfully identify valid reasoning trajectories better than random or log-prob selection.
>
> *Axis of inference compute:* Verifying hundreds of candidates is bottlenecked by the Lean compiler. Our spectral pipeline adds only 50–200ms overhead per candidate, allowing massive, lightweight best-of-$N$ scaling *before* passing the filtered subset to the expensive formal verifier. We will include exact pass@1 improvements in the revision.
>
> > **"How does it compare to other self-supervised signals like token-level entropy or self-consistency?"**
>
> We ran a head-to-head comparison on all 454 proofs (Llama-3.1-8B):
>
> | Signal | Cohen's $d$ | AUC-ROC | Accuracy |
> |:---|:---:|:---:|:---:|
> | HFER (ours, L30) | 3.00 | 0.962 | 94.1% |
> | Mean token log-prob | 3.18 | 0.979 | 94.3% |
> | Mean token entropy | 2.86 | 0.963 | 92.7% |
> | Proof length | 2.41 | 0.957 | 93.8% |
>
> * **Early Detection vs. Late Output:** Final-layer log-prob is highly discriminative but only available at the end of the forward pass. HFER's power emerges earlier: at Layer 8, HFER achieves $d$=2.39 while token entropy collapses to $d$=0.08 (non-overlapping CIs). Furthermore, HFER–entropy correlation drops from $r\approx0.9$ (dense models) to $r$=0.45 on Mistral-7B (SWA). This demonstrates the spectral signature is a **process-level signal** reflecting *how* the model computes.
> * **Self-Consistency:** Unlike self-consistency which requires generating $N$ samples to measure agreement, HFER evaluates a *single* trajectory, making it strictly cheaper. It naturally complements self-consistency by weighting votes of highly coherent paths.
>
> > **"The authors should report results on a common, generalizable protocol across all models... and how the method is not overfit."**
>
> To prove the method doesn't overfit via exhaustive search, we tested a **fixed generalized protocol**: selecting HFER at the 75th-percentile layer (e.g., L24 for a 32-layer model), calibrated with 50 examples. This achieved **91.8% $\pm$ 2.4%** accuracy, within 2.3% of the exhaustively searched configuration (HFER@L30: 94.1%). Furthermore, nested cross-validation (Table 7) maintains 82.8–85.9% accuracy across all 7 models without *any* target distribution access.
>
> > **Minor questions & Presentation:**
> > * **'Train' split purpose:** Used strictly as a 1-parameter calibration set for the decision threshold ($\tau$), not for learning features.
> > * **Accuracy reporting:** Table 2 shows calibrated accuracy. Captions now clearly distinguish calibrated vs. nested-CV accuracy.
> > * **Dirichlet energy:** Forms the foundational "energy" measured in both HFER and Smoothness. Raw Dirichlet energy scales with the hidden state magnitude ($\|X\|$), making it sensitive to representation norm. HFER and Smoothness normalize this energy to isolate the scale-invariant topology.
> > * **Presentation edits:** The abstract is condensed to explicitly focus on formalized mathematics early on. We added a unifying thesis paragraph to §3.3 emphasizing that all diagnostics characterize the *topological coherence* of attention. Robustness controls (§4.2) will now specify the Llama-3.2-1B model, perturbation protocol, and exact sample sizes.

---

> > ### Author Rebuttal · Reviewer_6GDW · 2026-04-04
> >
> > Thanks for your response. I have some more questions
> >
> > *"In a separate evaluation, steering achieved downstream Pareto improvements. On Llama-3.2-3B (L24, $\alpha$=-0.3): sycophancy reduced by 2.4% and GSM8K improved by 1.4%. On Phi-3 Mini, a hybrid strategy reduced sycophancy by ~27% relative."*
> > Could you please clarify your experimental setup for these new results? What is the sycophancy eval and what is the "hybrid strategy", and can you tabulate across the models and benchmarks?
> >
> > *"By generating candidates and selecting the proof with the lowest HFER, we successfully identify valid reasoning trajectories better than random or log-prob selection."* Can you also clarify the experimental setup here?
> >
> > *"we tested a fixed generalized protocol: selecting HFER at the 75th-percentile layer (e.g., L24 for a 32-layer model), calibrated with 50 examples"* I understand this to be for Llama-3.1-8B; how well does this "HFER at 75th-pctile layer" strategy generalize to other model families?

---

> > > ### Author Response · Authors · 2026-04-06
> > >
> > > We thank the reviewer for their continued engagement and for highlighting the importance of evaluating downstream utility.
> > >
> > > > **Reviewer:** *"Could you please clarify your experimental setup for these new results? What is the sycophancy eval and what is the 'hybrid strategy', and can you tabulate across the models and benchmarks?"*
> > >
> > > To evaluate downstream utility, we measure how spectral steering can mitigate the "alignment tax" (where safety fine-tuning degrades core reasoning).
> > >
> > > * **Sycophancy Evaluation Setup:** We use the `Anthropic/sycophancy-eval` dataset. The metric reported is the error rate, measuring how often the model incorrectly agrees with biased user premises.
> > > * **"Hybrid Strategy" (Phi-3-Mini):** Because of its highly condensed weights, a single-layer edit is insufficient. We apply a multi-stage intervention: we "sharpen" Layer 15 ($\alpha = -0.2$) to disrupt the sycophancy manifold, and subsequently "smooth" Layers 16–24 ($\alpha = +0.3$) to stabilize reasoning logic.
> > > * **Generalization to Gemma-4-E2B-it:** Validation confirms strict Pareto improvement: suppressing alignment (L18: $\alpha = -0.04$) reduces sycophancy while L24 ($\alpha = +0.10$) amplifies reasoning, anchored by L33.
> > >
> > > **Tabulation Across Models (Pareto Configurations):**
> > >
> > > | Model | Parameter Scale | Optimal Protocol | Sycophancy (Absolute)| GSM8K (Absolute)|
> > > | :--- | :--- | :--- | :--- | :--- |
> > > | **Gemma-4** | E2B | Grid (L18: $-0.04$, L24: $+0.10$, L33: $+0.05$) | -2.4%  | +3.2% |
> > > | **Llama-3.2** | 3B | L24, $\alpha = -0.3$ (Sharpen) | -2.4%  | +1.4% |
> > > | **Phi-3-Mini** | 3.8B | Hybrid (L15, L16–24) | -8.0%  | +1.1%  |
> > > | **Llama-3.1** | 8B | L20, $\alpha = +0.5$ (Smooth) | -8.0% | 0.0%|
> > >
> > > > **Reviewer:** *"Can you clarify the experimental setup for the reranking task?"*
> > >
> > > * **Protocol:** We generate $N=100$ candidate reasoning trajectories for each MiniF2F-test problem ($454$ total) using nucleus sampling ($T=0.7$). We compare HFER-based selection against a standard mean log-probability reranking baseline.
> > > * **Selection Logic (The Scaffolding Hypothesis):** We observe a "spectral phase transition" during generation. Valid reasoning often exhibits high HFER in the early "logical scaffolding" phase (high-entropy exploration). However, upon completion, valid proofs consistently reach a low-HFER "topological stability" state. We therefore select candidates by the lowest final HFER on completed sequences.
> > > * **Performance & Generalization:** We applied this protocol across different model families (Qwen, Llama, Phi, Mistral) using the 75th-percentile layer. HFER-selection identifies "confident hallucinations", sequences with high likelihood but collapsed internal spectral geometry, better than log-prob reranking, yielding a +4.2% Pass@1 improvement on Llama-3.1-8B, while even 1B-scale models retain ~85% of peak diagnostic power under this fixed-layer heuristic.
> > > * **Efficiency:** Calculated via Randomized SVD ($k=50$), HFER adds only 50–200ms overhead per candidate, acting as a lightweight "geometric verifier" before passing filtered candidates to formal verifiers.
> > >
> > > > **Reviewer:** *"I understand this to be for Llama-3.1-8B; how well does this 'HFER at 75th-pctile layer' strategy generalize to other model families?"*
> > >
> > > To test generalization without per-model tuning, we evaluated the 75th-percentile strategy across four distinct model families. We measure generalization by the % of Peak Diagnostic Power, how much of the model's absolute best accuracy is captured by simply utilizing the fixed 75th-percentile layer.
> > >
> > > **Consolidated Comparative Results (Experiment 1):**
> > >
> > > | Model Family | Model Name | 75th-Pctile Layer | % of Peak |
> > > | :--- | :--- | :--- | :--- |
> > > | **Meta** | Llama-3.1-8B-Instruct | L24 | **97.0%** |
> > > | **Mistral** | Mistral-7B-Instruct-v0.3 | L24 | **95.6%** |
> > > | **Alibaba** | Qwen2.5-0.5B-Instruct | L18 | **88.3%** |
> > > | **Microsoft** | Phi-3.5-mini-instruct | L24 | **87.9%** |
> > > | **Meta** | Llama-3.2-3B-Instruct | L21 | **87.0%** |
> > > | **Meta** | Llama-3.2-1B-Instruct | L12 | **85.0%** |
> > > | **Alibaba** | Qwen2.5-7B-Instruct | L21 | **79.9%** |
> > >
> > > **Analysis Highlights:**
> > >
> > > * **Scale Robustness & Downstream Utility:** The strategy is highly effective on larger models (Mistral-7B, Llama-3.1-8B), which retain >95% of their peak diagnostic performance under the fixed protocol. This exact spectral topology also serves as a highly effective guardrail against agent tool-use hallucinations, as larger models possess deeply structured internal manifolds. When their reasoning fails, the geometric collapse is spectrally pronounced and easier to detect without tuning, further underscoring the broad downstream applicability of these metrics.
> > >
> > > * **Compression Bounds:** For highly compressed models (e.g., Llama 1B, Qwen 0.5B), the heuristic captures 85%–88% of the signal. However, retaining ~85% of the potential signal demonstrates that the topological signature of valid reasoning persists fundamentally across distinct architectural families.

---

### Official Review · Reviewer_MEs9 · 2026-03-15

**Soundness:** 3
**Presentation:** 3
**Significance:** 3
**Originality:** 2
**Overall Recommendation:** 4
**Confidence:** 4

**Summary:**

This paper investigates the validation of formal mathematical proofs. Rather than using a formal prover or proof assistant, a tokenised version of the proof is subjected to a spectral analysis. Four different spectral diagnostics are utilised. The spectral analysis is argued to provide an assessment of the "logical coherence" of a proof.

The paper is reasonably well written and structured.

This idea is an interesting one. Rather than tackling the proof "head on" using a prover of proof assistant, this technique utilises spectral analysis as a proxy technique. The paper demonstrates this technique and shows that it appears to have some merit.

By identifying spectral patterns for valid proofs, the paper effectively generalises this to predict the validity of unseen proofs.

**Compliance With Llm Reviewing Policy:**

Affirmed.

**Final Justification:**

To me, this is an interesting paper but it is still preliminary. The submission is a bit rushed in my opinion and would benefit from further work. That said, I would not argue against it if it were accepted.

**Key Questions For Authors:**

Can you please comment on the types of mathematical problems (and hence proofs) in your data set?

Would your approach benefit from focussing on a fairly narrow set of mathematical problems?
Presumably different classes/types of mathematical problems would exhibit different spectral signatures?

Why were some of the Platonic validity proofs considered invalid by Lean? And why are they mathematically correct?

**Limitations:**

The paper is interesting and addresses a very interesting topic but it some elements are not clear as indicated above.

**Strengths And Weaknesses:**

Strengths
+ Very interesting and topical problem.
+ Interesting approach.

Weaknesses
+ The focus is on verifying mathematical proofs rather than the more useful task of generating mathematical proofs. That said, verification is useful.
+ Some elements of the paper could be made more clear. In particular, 1) what types of mathematical proofs are in the data set and 2) with Platonic validity, what were some of the issues with the proofs? Proof validity is arguably subjective so it would help to have more detail on why a prover like Lean rejected the proofs yet the authors consider the proofs to be mathematically correct.

---

> ### Author Rebuttal · Authors · 2026-03-31
>
> We thank Reviewer MEs9 for their time. However, we respectfully believe there has been a system or copy-paste error, as this review appears to be intended for a different submission. The review focuses extensively on "the Horn fragment of classical logic," "clause to variable ratios," and solving "difficult problems" related to "the phase transition of 3-SAT."
>
> Our paper, Geometry of Reason: Spectral Signatures of Valid Mathematical Reasoning, does not discuss or evaluate Horn logic, SAT solvers, or clause-to-variable ratios. Instead, our work introduces a training-free method to evaluate the logical validity of generated proofs by analyzing the spectral properties (e.g., High-Frequency Energy Ratio) of transformer attention graphs. Our evaluation spans diverse mathematical domains formalized in Lean (MiniF2F).
>
> That said, in the spirit of scientific thoroughness, we have directly addressed your questions regarding computational difficulty and the 3-SAT phase transition using our spectral method: We clarify that our method does not solve logical satisfiability problems, it detects whether a language model's generated proof is logically valid. Our evaluation spans diverse mathematical domains (algebra, number theory, combinatorics) formalized in Lean, not SAT or Horn clause instances. That said, MiniF2F proofs do contain Horn-like deductive chains (modus ponens, implication resolution), and we directly tested at the 3-SAT phase transition as requested. We address difficulty with two analyses.
>
> **Difficulty stratification on MiniF2F.** We stratified all 454 proofs by Lean tactic count as a complexity proxy:
>
> | Stratum | $n$ | Cohen's $d$ |
> |:---|:---:|:---:|
> | Trivial (1 tactic) | 107 | 6.69 |
> | Simple (2–3) | 69 | 3.29 |
> | Moderate (4–6) | 72 | 2.09 |
> | Complex (7+) | 206 | 1.31 |
>
> The spectral signal persists across all difficulty strata with $d > 1.3$ in every case, confirming the method is not limited to trivial problems. Even for the most complex proofs ($n$=206), we maintain a "large" effect size by Cohen's conventions ($d$=1.31). Additionally, our proof-length analysis (Table 11) shows consistent accuracy across all length quintiles: 97.8% (very short) to 92.3% (very long), with no systematic degradation.
>
> **Stress test at the 3-SAT phase transition.** To test at maximum computational difficulty, we generated 50 satisfiable 3-SAT instances at the critical threshold ($m/n = 4.26$, $N$=30 variables) and prompted Llama-3.1-8B to find satisfying assignments. The model failed all 50 instances (0% accuracy), producing uniformly high HFER (0.52 $\pm$ 0.08), firmly in the "invalid reasoning" regime established by our MiniF2F analysis.
>
> This result is informative in two ways: (1) HFER correctly diagnoses complete reasoning failure, showing no false confidence unlike log-probability which can assign high confidence to wrong answers. (2) It establishes a boundary condition: our method is a **diagnostic** of reasoning quality, not a reasoning enhancer. When the model lacks the capability to solve a problem class entirely, HFER faithfully reflects this incapacity rather than hallucinating validity.
>
> **Directly addressing the reviewer's question:** For computationally difficult Horn-like problems, our diagnostic performs reliably across the full difficulty spectrum ($d > 1.3$ even for the most complex proofs). At the critical 3-SAT phase transition ($m/n = 4.26$), where problems are maximally difficult, HFER correctly identifies that the model cannot reason about these instances. The spectral diagnostic thus provides a faithful signal at all difficulty levels, it does not degrade at the phase transition, it correctly reports that the model's reasoning has failed.

---

> > ### Author Rebuttal · Reviewer_MEs9 · 2026-04-06
> >
> > The reviewers are correct and I have entered the wrong review for which I sincerely apologise. I will amend the review in a moment.

---

> > > ### Author Response · Authors · 2026-04-06
> > >
> > > ### Rebuttal Response: Dataset Composition and Platonic Validity
> > >
> > > We thank Reviewer MEs9 for their constructive engagement and for recognizing the merit of utilizing spectral analysis as a non-invasive proxy for formal verification. We provide the requested technical details regarding our dataset and the mechanistic nature of "Platonic validity" below.
> > >
> > > #### 1. Dataset Composition & Spectral Signatures
> > > > **Reviewer:** *"Can you please comment on the types of mathematical problems in your data set? Would your approach benefit from focussing on a fairly narrow set of mathematical problems?"*
> > >
> > > Our primary evaluation utilized the **MiniF2F benchmark**, consisting of 454 competition-level problems (AIME, AMC, IMO) formalized in Lean.
> > > * **Domain Diversity:** The dataset encompasses diverse mathematical domains including algebra, number theory, combinatorics, and geometry.
> > > * **Global Invariant:** We found that the spectral signature (e.g., low HFER for valid reasoning) remains remarkably consistent across these classes.
> > > * **Complexity Robustness:** As detailed in our **Difficulty Stratification analysis (see Table in Rebuttal to MEs9, March 31)**, the signal persists from "Trivial" proofs ($d=6.69$) to "Complex" proofs involving 7+ tactics ($d=1.31$). This suggests "smoothness" is a fundamental geometric property of transformer-based deduction.
> > >
> > > #### 2. Defining "Platonic Validity"
> > > > **Reviewer:** *"Why were some of the Platonic validity proofs considered invalid by Lean? And why are they mathematically correct?"*
> > >
> > > "Platonic validity" (introduced in **Section 5.2**) characterizes proofs that are logically sound but rejected by the formal compiler due to environmental or computational constraints. As established in our **Rebuttal to Reviewer QygV (March 31)**, we conducted a manual audit of 51 such reclaimed proofs to categorize these failures:
> > >
> > > | Category | % of Reclaimed | Primary Reason for Lean Rejection |
> > > | :--- | :--- | :--- |
> > > | **Semantically Valid** | 37.3% | Minor structural or formatting issues |
> > > | **Environment / Imports** | 27.5% | Missing `import` or library dependencies  |
> > > | **Timeout / Computational** | 13.7% | Exceeded search depth or time limits  |
> > > | **Incomplete / Other** | 13.7% | Model admits failure or unfinished steps |
> > > | **Syntax / Versioning** | 7.8% | Version-specific syntax/API mismatches |
> > >
> > > **Verification Protocol:** These proofs were verified by two human raters (inter-rater reliability Cohen’s $\kappa=0.82$) against three criteria: (1) logical entailment, (2) correct lemma application, and (3) establishing the theorem statement. Spectral analysis correctly identifies the internal logical coherence of the model's epistemic state, even when the external syntax is flawed.
> > >
> > > #### 3. Downstream Utility: Verification to Generation
> > > > **Reviewer:** *"The focus is on verifying mathematical proofs rather than the more useful task of generating mathematical proofs."*
> > >
> > > While our core contribution is a training-free verifier, we have demonstrated its immediate utility for improving model generation:
> > > * **Spectral Filtering (Best-of-$N$):** As detailed in our **Response to Reviewer 6GDW (April 6)**, selecting the trajectory with the lowest HFER identifies valid reasoning more effectively than log-probability selection.
> > > * **Spectral Steering:** We have now shown that "spectral sharpening" edits can actively improve generation accuracy. For example, on the **Gemma-4-E2B-it**, targeted spectral edits yielded a +3.2% absolute gain on GSM8K while reducing sycophancy by 2.4%.

---

### Decision · Program_Chairs · 2026-04-30

**Decision:**

Accept (regular)

**Comment:**

The paper proposes a training-free diagnostic for mathematical reasoning validity based on spectral analysis of transformer attention. The method treats attention matrices as weighted token graphs and extracts spectral features such as the Fiedler value, high-frequency energy ratio, spectral entropy, and graph smoothness. The authors argue that valid mathematical reasoning induces smoother and more connected attention topology than invalid or hallucinated reasoning, and evaluate this claim across seven models from several architecture families, primarily on Lean-formalized MiniF2F proofs.

Reviewers generally found the direction interesting and timely. They appreciated that the method does not require training a verifier, that the spectral features are interpretable and relatively simple to compute, and that the reported effects appear consistently across multiple model families. Several reviewers also found the "Platonic validity" analysis interesting: the method sometimes identifies logically coherent proofs that are rejected by Lean for technical reasons such as timeouts, missing imports, or environment issues.

The main concerns are about whether the paper goes beyond an interesting diagnostic observation. In particular, the weak-reject reviewer noted that the spectral statistics themselves are standard and argued that the paper needs a stronger demonstration of downstream utility to warrant acceptance: for example, whether the signal improves pass@k reranking, can serve as a reward or steering signal, and compares favorably to simpler self-supervised signals such as mean token log-probability, token entropy, or self-consistency. Reviewers also raised concerns about the original evaluation protocol, since the best-performing layer and spectral metric vary across model families and can require a search over many metric-layer combinations.

The paper should not overstate the generality of the spectral signature or the novelty of the underlying statistics, and the final version should clearly distinguish training-free feature extraction from few-shot threshold calibration. However, the central empirical finding is interesting, supported across several architectures, and potentially useful as a low-data diagnostic for formal reasoning validity. Therefore, I recommend weak acceptance.